**COMMUNICATIONS**

# Heterogeneity within and among co-occurring foundation species increases biodiversity

Mads S. Thomsen[1,2], Andrew H. Altieri[3,4], Christine Angelini[4], Melanie J. Bishop[5], Fabio Bulleri[6], Roxanne Farhan[7], Viktoria M. M. Frühling[3], Paul E. Gribben [8,9], Seamus B. Harrison[3], Qiang He [10 ✉], Moritz Klinghardt[11], Joachim Langeneck[6], Brendan S. Lanham [8,9], Luca Mondardini[1], Yannick Mulders[12], Semonn Oleksyn[5], Aaron P. Ramus [13], David R. Schiel[1], Tristan Schneider[11], Alfonso Siciliano[1], Brian R. Silliman[14], Dan A. Smale [15], Paul M. South[16], Thomas Wernberg [12], Stacy Zhang [14] & Gerhard Zotz[3,11]

Habitat heterogeneity is considered a primary causal driver underpinning patterns of diversity, yet the universal role of heterogeneity in structuring biodiversity is unclear due to a lack of coordinated experiments testing its effects across geographic scales and habitat types. Furthermore, key species interactions that can enhance heterogeneity, such as facilitation cascades of foundation species, have been largely overlooked in general biodiversity models. Here, we performed 22 geographically distributed experiments in different ecosystems and biogeographical regions to assess the extent to which variation in biodiversity is explained by three axes of habitat heterogeneity: the amount of habitat, its morphological complexity, and capacity to provide ecological resources (e.g. food) within and between co-occurring foundation species. We show that positive and additive effects across the three axes of heterogeneity are common, providing a compelling mechanistic insight into the universal importance of habitat heterogeneity in promoting biodiversity via cascades of facilitative interactions. Because many aspects of habitat heterogeneity can be controlled through restoration and management interventions, our findings are directly relevant to biodiversity conservation.

[1] Marine Ecology Research Group and Centre for Integrative Ecology, School of Biological Sciences, University of Canterbury, Christchurch, New Zealand. [2] Department of Bioscience, Aarhus University, 4000 Roskilde, Denmark. [3] Smithsonian Tropical Research Institute, Apartado, 0843-03092 Balboa, Ancon, Republic of Panama. [4] Environmental Engineering Sciences, University of Florida, Gainesville, FL, USA. [5] Department of Biological Sciences, Macquarie University, Sydney, NSW, Australia. [6] Dipartimento di Biologia, Università di Pisa, CoNISMa, Via Derna 1, 56126 Pisa, Italy. [7] Marine Sciences, University of Georgia, Athens, GA, USA. [8] Centre for Marine Science and Innovation, School of Biological, Earth and Environmental Sciences, University of New South Wales, Sydney, NSW, Australia. [9] Sydney Institute of Marine Science, Chowder Bay Road, Mosman, 2088 Sydney, NSW, Australia. [10] Coastal Ecology Lab, MOE Key Laboratory for Biodiversity Science and Ecological Engineering, School of Life Sciences, Fudan University, 2005 Songhu Road, 200438 Shanghai, China. [11] Institute for Biology and Environmental Sciences, Carl von Ossietzky University Oldenburg, Oldenburg, Germany. [12] School of Biological Sciences and UWA Oceans Institute, University of Western Australia, Perth, WA, Australia. [13] Department of Biology and Marine Biology, University of North Carolina Wilmington, Wilmington, NC, USA. [14] Nicholas School of the Environment, Duke University, 135 Duke Marine Lab Road, Beaufort, NC, USA. [15] Marine Biological Association of the United Kingdom, The Laboratory, Citadel Hill, Plymouth PL1 2PB, UK. [16] Cawthron Institute, Nelson, New Zealand. ✉email: He_Qiang@hotmail.com

Understanding the processes that determine patterns of biodiversity is a cornerstone of ecology, evolution, biogeography and conservation biology. Many models and mechanisms can account for patterns of biodiversity, including disturbance regimes, ecosystem age, stability, climatic variability, productivity, energy levels, island sizes, isolation and direct and indirect species-interactions[1–9]. Underpinning many of these models and mechanisms is the idea that habitat heterogeneity and complexity—umbrella terms for all aspects of habitat variability, including the amount, function, and form and shape of habitat features[7]—increases biodiversity of associated species by increasing the quantity, quality and breadth of resources, refugia and niche spaces[7,10–14]. Studies on habitat heterogeneity range from detailed measurements of compositional and configurational variability in habitat characteristics to broader emphases on nonuniformity in a wide variety of habitat traits[2,7,11,15–17]. Here we adopt a broad approach where heterogeneity and complexity are considered equivalents[2,16], that can be measured from traits such as the arrangement, diversity and function and sizes and abundances of structural elements[2,16]. For example, seminal work by MacArthur and MacArthur[18] demonstrated how bird diversity increases with 'foliage height diversity' (diversity of elements), an index that varies with the distribution of vegetation in space, including the width and height of vegetation strata (sizes of elements). Similarly, Tews et al.[16] suggest that the sizes and abundances of keystone structures increase habitat heterogeneity and biodiversity, equivalent to our analyses of biodiversity associated with epiphytes attached to seaweed and seagrass and bivalves embedded within marshes and mangroves (Fig. 1). Although most studies have shown positive relationships between heterogeneity and biodiversity, certain processes, like effective area per species, can result in negative relationships between heterogeneity and biodiversity[15,19].

Relationships between habitat heterogeneity and biodiversity may be driven by processes along at least three broad axes of variation: the amount of the habitat, the function of the habitat and the form and shape of habitats (hereafter habitat morphology). First, where heterogeneity measurements are based on variation in the amounts of structural elements, positive relationships between heterogeneity and biodiversity reflect classic patch theory, by which greater amounts of habitat support both more species and larger populations that are more resilient to disturbances[2,9,16,20]. Second, heterogeneity associated with habitat function emphasises that different species use specific habitats to acquire specific resources. Habitats that provide many ecological functions, such as different physical structures, hiding, nesting and resting places, nutritional resources, and geochemical and nutritional pathways, should therefore support more species than similar habitats that provide fewer functions[14,16,21]. For example, only forests with trees of specific species, sizes, ages and cavities are inhabited by nesting woodpeckers[22] and only forests with a variety of tree-hosts and mistletoes are inhabited by specialist feeders like mistletoe birds[23]. In other words, a forest is only fully functional for woodpecker and mistletoe birds if it is heterogeneous and supports all habitat-requirements related to successful feeding, breeding, nesting, resting and hiding. A third axis of heterogeneity is habitat morphology, whereby landscapes with more diverse, complex and irregular shapes of habitat-generating elements provide more niche space, refugia from enemies and specialised habitat, and therefore support higher biodiversity[2] (sometimes referred to as compositional heterogeneity[15]).

The importance of each axis has been assessed with observational data (which may be biased by uncontrolled confounding factors) and modelling[7,10–12,14,15]. However, no studies have assessed, with controlled experiments, whether these axes of heterogeneity have additive, synergistic or antagonistic interactive effects, their relative importance, or their generality across habitats, ecosystems, biogeographical regions and taxonomic resolution.

Each of the three axes of heterogeneity can be created, modified or controlled by particular organisms. For example, primary foundation species (FS), like trees, saltmarsh plants, and kelp, increase biodiversity by providing living space, food and shelter for animal species[24]. Often, primary FS also provide habitat to secondary FS, such as mistletoes, epiphytes or oysters, and these secondary FS can increase local habitat-associated heterogeneity by adding new and different resources, refugia and niches through facilitation cascades[23,25–31]. Mirroring early trophic cascade studies[32], facilitation cascade research has focused on testing how individual factors, such as plant type and animal body size, control facilitation cascades[26,28,31,33] and documenting generality across ecosystem types[34]. However, as is the case with trophic cascades, many interacting factors and drivers can modify and control facilitation cascades. We propose that many system-specific drivers could be accounted for by the three axes of habitat heterogeneity, with predictable and general relationships that explain biodiversity across ecosystem types.

Specifically, we posit that these axes can determine biodiversity patterns by controlling facilitation cascades. First, we predict that greater amounts of habitat generated by secondary FS will increase biodiversity in a facilitation cascade through creation of more habitat that can support larger populations of associated animals, and by increasing taxonomic richness (i.e. species density[35]) through a sampling effect[27,28,36]. Second, we expect that biodiversity in facilitation cascades will be higher when secondary FS add more resources—i.e. more ecological functions—to support different species, relative to the primary FS alone. This axis of heterogeneity has, in the past, been tested experimentally by comparing animal biodiversity associated with living FS that provide food for consumers, alter environments through metabolic activities, and add organic material to decomposer webs and refugia for animals to escape enemies to similar looking non-living mimics of FS that only provide a habitat refuge from enemies or adverse environmental conditions[31,37–39]. Finally, we anticipate that biodiversity in facilitation cascades will increase when secondary FS are morphologically distinct from primary FS (here operationalized as 'Δmorphology' where values >0 implies that the secondary FS is morphologically more complex, branched, and convoluted compared to the primary FS, see Methods for details). For example, if secondary FS are morphologically more convoluted than primary FS, and provide more, as well as different, interstitial spaces and micro-habitats than primary FS, then the facilitation cascades should support more size-classes, larger local populations and greater taxonomic richness[2,27,36,40].

To examine these effects and their generality, we used geographically distributed experiments, which have been heralded for their ability to test for consistency in mechanistic processes and associated biodiversity patterns across habitats, ecosystems and biogeographical regions[41,42]. This approach has so far focused on individual processes in specific habitats, such as biodiversity-productivity relationships in grasslands[43], nutrient impacts on seagrasses[44] or warming effects in tundra[45]. Here, we adapted this approach to test for, and rank in importance, three ecological processes that could simultaneously affect how facilitation cascades control biodiversity across habitats, ecosystems and regions[46]. Specifically, we tested whether biodiversity (measured in terms of animal abundances, taxonomic richness and multivariate community structure) is affected by the three axes of habitat heterogeneity in facilitation cascades: the amount, function and morphological differences among FS[2,7]. We focused on animal, not plant, responses, to follow the standard approach of

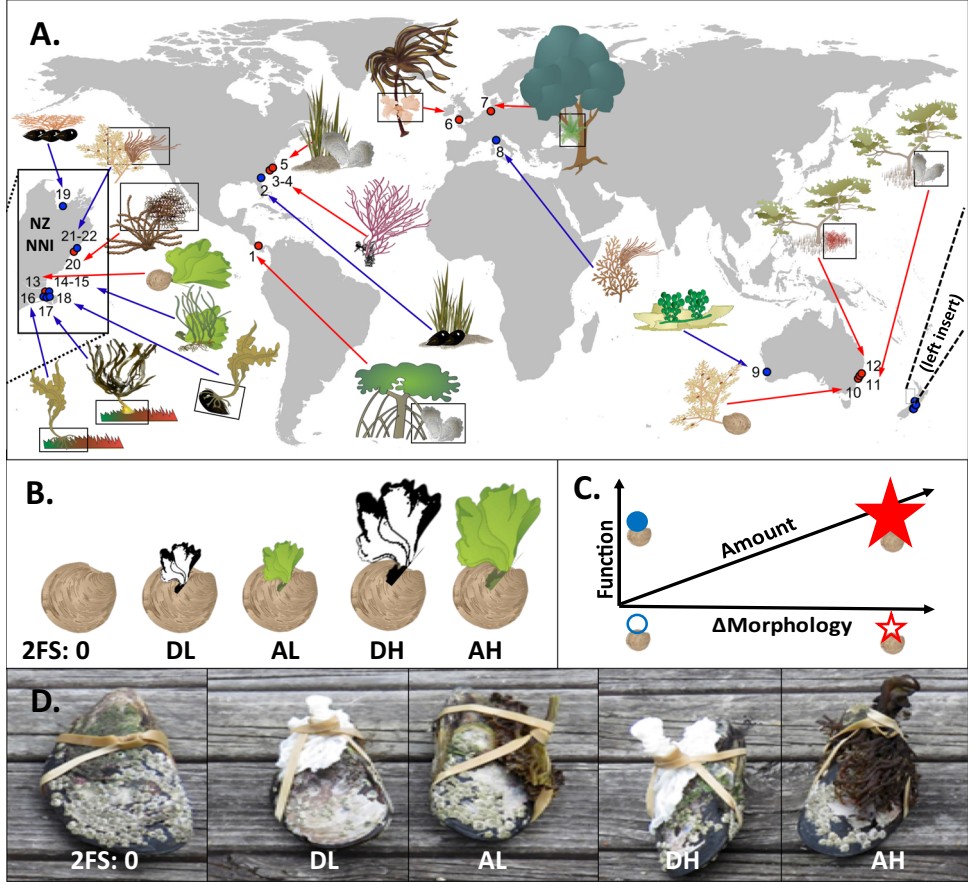

**Fig. 1 Locations and experimental design of 22 geographically distributed facilitation experiments testing for effects of habitat heterogeneity on biodiversity. A** Map showing the distribution of 22 analogous facilitation cascade experiments numbered from west to east beginning in North America. The insert map on the left is a close-up of New Zealand Northern South Island (NSI NZ). **B** Schematic diagram of experiment 13 shows the control where the primary foundation species (FS), the cockle *Austrovenus*, is alone and four treatments where the cockle co-occur with secondary foundation species (the seaweed *Ulva*) that are dead (or mimics) (D) or alive (A) in low (L) or high (H) amounts. The 22 experiments were grouped into 11 where the secondary FS was morphologically more complex than the primary FS (red line) and 11 where the morphology was comparable or less complex than the primary FS (blue lines) (=Δmorphology test-factor). **C** Schematic diagram showing three orthogonal axes of habitat-associated heterogeneity (=crossed test-factors) with two amounts (low = small vs. high = large symbols), two levels of ecological functions (dead = open vs. alive = closed symbols) and two Δmorphologies (low = blue circle vs. high = red star) of the secondary FS. **D** Photos showing out-transplanted controls and treatments for experiment 18. Experiment 3–4, 14–15 and 21–22 were done in two seasons to examine temporal effects. Black squares in 1 A = experiments where only a part of the primary FS was sampled, such as prop roots, pneumatophores, tree branches, kelp stipes, and kelp holdfasts. Genera involved in the experiments were (primary FS → secondary FS): (1) *Rhizophora* prop root → *Magallana*, (2) *Spartina* → *Geukensia*, (3–4) *Diopatra* tube mimic → *Gracilaria*, (5) *Spartina* → *Crassostrea*, (6) *Laminaria* stipe → *Palmaria*, (7) *Fagus* branch → *Polypodium*, (8) *Halopithys* → *Jania*, (9) *Pseudoceratina* mimic → *Caulerpa*, (10) *Anadara* → *Sirophysalis*, (11) *Avicennia* pneumatophore → *Saccostrea*, (12) *Avicennia* pneumatophore → *Bostrychia/Caloglossa*, (13) *Austrovenus* → *Gracilaria*, (14-15) *Zostera* → *Ulva*, (16) Turf algal mimic → *Undaria* holdfast, (17) Turf alga; mimic → *Durvillaea* holdfast, (18) *Perna* → *Undaria* holdfast, (19) *Xenostrobus* → *Capreolia*, (20) *Hormosira* → *Notheia*, (21-22) *Cystophora* → *Polysiphonia*. The kelp figures from experiment 6 and 17 are our own and the remaining plant and animal figures are from Integration and Application Network (ian.umces.edu/media-library).

past facilitation cascade research[8] (but see[47] for an exception). We performed 22 geographically distributed factorial field experiments (Fig. 1, Supplementary Table 1) testing whether biodiversity in facilitation cascades is higher when: (1) the secondary FS is present in greater amounts, resulting in more resources and therefore higher colonisation by more individuals and species[27,28,36]; (2) the secondary FS is alive and provide many ecological functions, such as trophic subsidies, waste products for decomposers, biogeochemical fluxes, stress-amelioration and habitat space, compared to morphologically similar non-living structures that only provide stress-amelioration and habitat space[31,37–39]; and (3) the secondary FS is morphologically more complex, branched and convoluted than the primary FS and thereby forms more niches and specialised microhabitats[27,36]. From these experimental data we show that biodiversity increases with the amount of biogenic habitat provided by secondary FS as well as its morphological complexity and capacity to provide ecological resources, and that positive effects are generally additive between test-factors. The results highlight the fundamental importance of habitat heterogeneity in promoting biodiversity via cascades of facilitative interactions.

## Results

Across the 22 experiments and different taxonomic resolutions, we found that the addition of secondary FS increased biodiversity measures (i.e. most Log response ratios were >0) and that biodiversity measures generally increased with increasing heterogeneity (Fig. 2A–F). Furthermore, communities associated with co-occurring FS increasingly diverged from communities

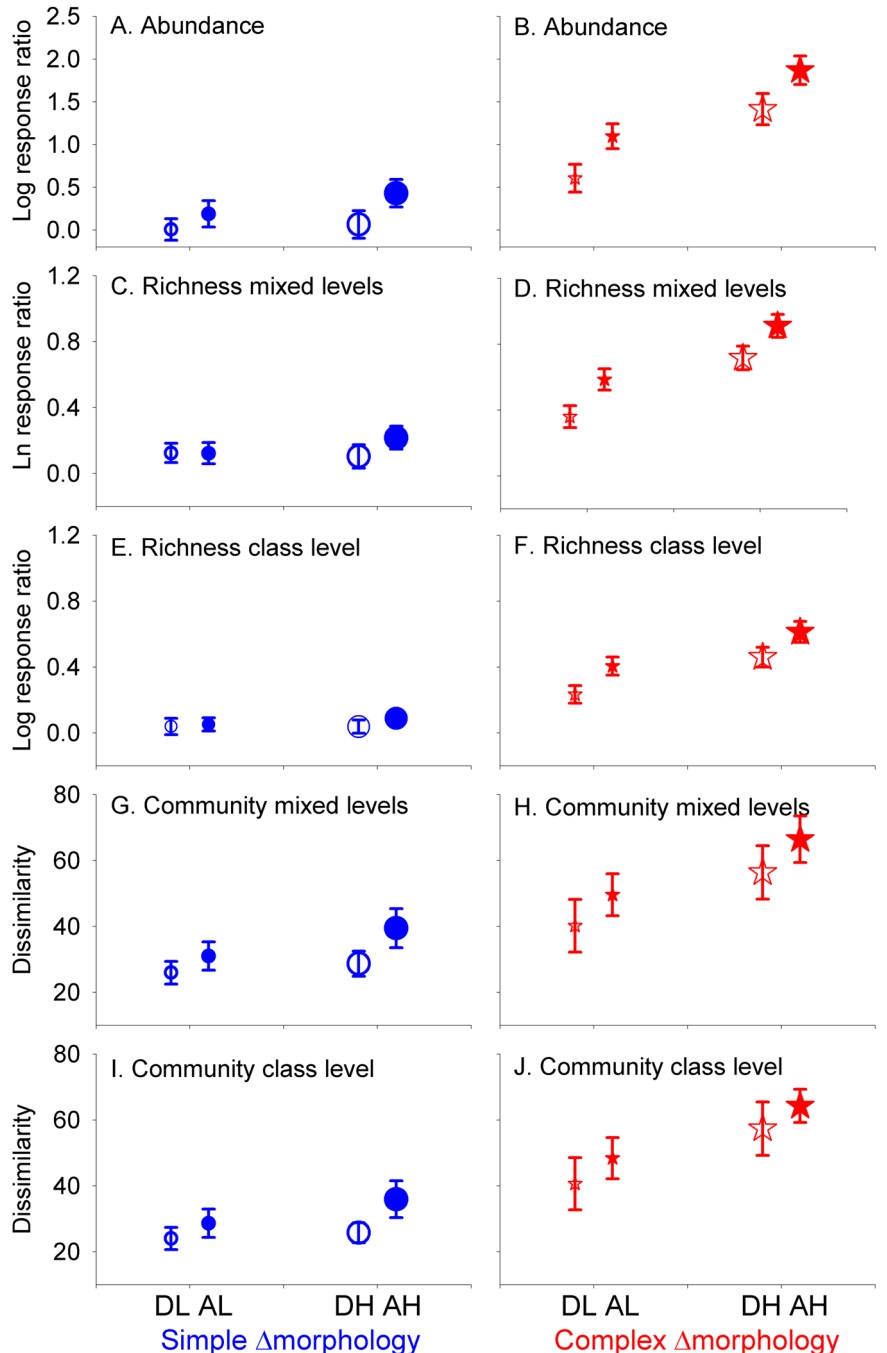

**Fig. 2 Effects of habitat heterogeneity on biodiversity from 22 geographically distributed facilitation experiments.** Effects of three orthogonal axes of habitat heterogeneity (Amount, Function, ΔMorphology) in 22 facilitation cascade experiments measured on animal abundances (**A**, **B**), taxonomic richness on mixed (**C**, **D**) and class (**E**, **F**) level, and Bray–Curtis community dissimilarity on mixed (**G**, **H**) and class (**I**, **J**) level. Data are presented as mean values ± 95% confidence limits. Log response ratios were calculated by comparing treatments where primary and secondary foundation species (FS) co-occur to controls where the primary FS was alone so that values above zero imply that the secondary FS increases biodiversity. Log response ratios were calculated from individual samples for abundance and richness (replication levels from left = 80, 75, 73, 78, 132, 135, 134, 128) whereas community dissimilarities were calculated for each combination of two sites and 11 experiments per Δmorphology (i.e. replication level = 22). Blue circles vs. red stars = Low vs. High ΔMorphology; small vs. large symbols = Low vs. High Amount; open vs. closed symbols = Low vs. High Functionality. Legend Text on figure: DL = Dead-Low amount, AL = Alive-Low amount, DH = Dead-High amount, AH = Alive-High amount. Data underpinning the figure are provided as online Source Data file.

associated with primary FS alone as the level of heterogeneity increased (Fig. 2G–J).

For abundance of associated organisms, we found significant interactions between amount and Δmorphology ($p < 0.001$, Sum of Squares = 20.31) and between amount and function

($p = 0.003$, Sum of Squares = 2.92), where positive effects of amounts were stronger when secondary FS had more convoluted morphologies compared to the primary FS and when secondary FS was alive (Fig. 2A–B). In addition, there were strong significant main effects for all three drivers of heterogeneity (Fig. 2A,

Supplementary Data 1), highlighting that animal abundance is greater at high amount ($p < 0.001$, Sum of Squares = 5.61), high functional heterogeneity ($p < 0.001$, Sum of Squares = 36.39) and high Δmorphology ($p < 0.001$, Sum of Squares = 6.60) of secondary FS relative to primary FS. Furthermore, there were significant positive effects of standardised biomass of both primary ($p < 0.001$, Sum of Squares = 2.01) and secondary ($p = 0.015$, Sum of Squares = 12.93) FS. Of the six spatiotemporal moderators, only start date ($p = 0.016$) and season ($p = 0.041$) significantly affected abundance, but with low Sum of Squares (2.25 and 1.46, respectively).

Results for taxonomic richness (Supplementary Data 1) were very similar to abundance results for both mixed (Fig. 2C–D) and class (Fig. 2E–F) level data, even though the Log response values were slightly larger on the mixed level. Again, we found significant interactions between amount and Δmorphology ($p < 0.001$, Sum of Squares = 4.41 and 1.81) and function and Δmorphology ($p = 0.005$ and 0.002, Sum of Squares = 0.78 and 0.66) demonstrating that the positive effects of amount and function were stronger when secondary FS had more convoluted morphologies compared to the primary FS (Fig. 2C–F). We also found strong significant main effects for the three heterogeneity drivers, again highlighting that richness was greater at high amount (statistics for mixed taxonomic levels is presented before class level; $p < 0.001$ and 0.002, Sum of Squares = 1.44 and 0.66), high functional heterogeneity ($p < 0.001$, Sum of Squares = 4.62 and 2.01) and high Δmorphology ($p < 0.001$, Sum of Squares = 3.11 and 1.48). There was also a positive significant effect of the standardised biomass of secondary FS ($p < 0.001$ and 0.026, Sum of Squares = 1.25 and 0.33). All other test factors were non-significant ($p > 0.13$, Sum of Squares < 0.3).

Results for community dissimilarity followed the same pattern (Fig. 2G–J, Supplementary Tables 2–4); the effect of amount was stronger when secondary FS had more convoluted and complex morphologies compared to the primary FS (amount × Δmorphology: $p = 0.009$ for both mixed and class level data, Sum of Squares = 1311 and 1521). Furthermore, dissimilarity was greater at high than at low amounts ($p < 0.001$, Sum of Squares = 5380 and 4774), functional heterogeneity ($p < 0.001$, Sum of Squares = 3421 and 2382) and Δmorphology ($p < 0.001$, Sum of Squares = 21040 and 25517) of secondary FS. Tests for all other factors were non-significant ($p > 0.44$, Sum of Squares < 116).

Comparing the explained Sum of Squares in the tests described above highlighted that, for abundance and richness data, heterogeneity (including biomass covariates) explained >93% of variation compared to <7% for spatiotemporal covariates (Supplementary Data 1). Furthermore, ranking the test-factors according to their Sum of Squares showed that for abundance and richness data, function was most important, followed by Amount × Δmorphology (the third ranked factor varied between tests). By comparisons, ranking dissimilarity data showed Δmorphology was most important, followed by amount and function (Supplementary Table 3).

## Discussion

Habitat-associated heterogeneity has been identified as a universal driver of biodiversity because heterogeneous environments are thought to have more resources, refugia and niche spaces[7,48–50]. This study of geographically distributed factorial field experiments found that the extent to which facilitation cascades promote animal abundances, taxonomic richness and differences in community structure increases with the magnitude by which secondary FS enhance heterogeneity over that provided by the primary FS alone. In general, we found positive and

additive effects of habitat amount, function and morphology (i.e. with only a few significant simple-to-interpret interactions) suggesting that secondary FS enhance biodiversity not only by increasing the amount of habitat space over that which was provided by primary FS alone, but also by providing protective microhabitats and other resources as living organisms, such as food for consumers and detritus.

There was strong support for our hypothesis that morphological differences between primary and secondary FS control the strength of facilitation cascades, as predicted by the results of a few case studies that compared facilitation cascades with different secondary FS[47,51–53]. Here, morphological heterogeneity was assessed by ranking primary and secondary FS as simple, intermediate, or complex shapes, a common approach in traditional form-functional ecology[54–56]. There also was strong support for our second hypothesis that living secondary FS, which can provide more resources (i.e. functions) to animals, create stronger facilitation cascades than non-living secondary FS. Living secondary FS likely promoted biodiversity because they are edible and produce detritus, thereby supporting both consumer and decomposer-based communities, and by modifying local conditions through metabolic process including gas exchange and excretion[23,38,57]. Nevertheless, even mimics of secondary FS had positive effects on biodiversity, highlighting the universal importance of habitat space in affecting biodiversity[2,40]. A combination of reduction in environmental stress (e.g. moisture retention beneath secondary FS in intertidal habitats) and the provision of predation refugia (e.g. in the interstitial spaces of secondary FS) likely explains facilitation by non-living secondary FS[34,37,38,58]. Future experimental studies should therefore explicitly test how effect sizes in facilitation cascades vary along stress and predation pressure gradients[34,59].

Finally, there were stronger facilitation cascades when the secondary FS was abundant or large, providing strong experimental support to the results of case studies suggesting that habitat-quantity controls facilitation cascades[26,28,31,33]. However, it is possible that extreme amounts of secondary FS may inhibit primary FS by increasing drag and breakages[60,61], competition for limited resources[62,63], parasitizing of hosts[60] or by creating adverse environmental conditions[64–66]. The greatest amounts of secondary FS used in our experiments were based on field observations to decrease the likelihood that we would exceed such a threshold. Mechanistically, having greater amounts of secondary FS increases habitat space, refugia and (for living secondary FS) food and can therefore increase inhabitant abundances and taxonomic richness, in part through stochastic sampling and patch-size effects[9,67,68]. Again, secondary FS had generally positive effects on diversity, even when the amount of secondary FS was reduced by 75% relative to high abundance treatments, demonstrating that the mere presence of a secondary FS can increase biodiversity.

To this point we have discussed axes of heterogeneity individually, but it is a major goal in ecology to understand how multiple factors act together to affect biodiversity[69–71]. For example, interacting factors can be non-additive and near-impossible to predict from single factor tests resulting in inadequate projections of future biodiversity[72,73]. In contrast to most studies on heterogeneity vs. biodiversity[11,15] or facilitation cascades[23,25–31], our factorial approach allowed us to test for such interaction effects. Importantly, synergism occurred between the amount and morphology of habitat of secondary FS (in all biodiversity responses), and between the amount and function of secondary FS (in abundances and richness responses), highlighting how the effects of high amounts of secondary FS are stronger when secondary FS are more convoluted or provide more functions (Fig. 2). These results suggest that limiting

facilitation thresholds exist along some heterogeneity axes, but that the thresholds can be exceeded when secondary FS become abundant[71]. Results of our global mechanistic experiments support many observational studies documenting positive relationships between heterogeneity and biodiversity but contrast a few studies that have shown unimodal or negative relationships for individual taxa or trophic groups[7,10–14,74]. Unimodal and negative relationships have been explained by specific combinations of habitat compositions and configurations, and animal niche-breadth and dispersal traits[15,74]. We probably found positive relationships because our heterogeneity tests were based on short binary treatments (instead of long continuous gradients[11]), dispersal is less limiting in small-scale facilitation cascade experiments where animals often move between primary and secondary FS[15], and taxa were grouped into one-dimensional community abundance and taxonomic richness values so that strong and common species-specific positive relationships overshadow minor weak or infrequent negative relationships[11]. Our factorial, experimental approach also allowed us to rank the axes of habitat-heterogeneity in order of importance (here based on sums of squares)[75]. Thus, for abundances and taxonomic richness, function was most important followed by $\Delta$morphology × amount (the third most important factor varied between tests), whereas the rankings for community structure was $\Delta$morphology followed by amount and function (consistent between tests). Perhaps this difference reflects that the two former responses are simple univariate metrics whereas the latter incorporates all species-specific data in multidimensional space[76]. To our knowledge this is the first facilitation cascade study that has explicitly ranked different forms of habitat-associated heterogeneity, although a few case studies have previously suggested that the strength of facilitation cascades can be modified by the presence of tertiary FS[53], habitat types[77,78] and densities of FS[28,51]. Altogether, this work highlights that the strength of facilitation cascades can be highly response- and context-dependent[79] as is the case for trophic cascades[32,80].

Many tests of global biodiversity patterns only assess taxonomic richness[1–7,9], i.e. they omit information about species identities and abundances and, therefore, properties that are of fundamental importance in selection and niche models[81,82]. However, biodiversity, estimated from the entire species-sample matrices, is fundamentally multivariate and density-dependent, and should therefore also be analysed with metrics that incorporate these properties[76,83,84]. Here, we found very similar results between univariate richness and abundance metrics and multivariate dissimilarity (a metric related to community turnover and beta-diversity as measured in traditional ecology, landscape ecology, biogeography and conservation biology[85–89]) revealing that facilitation cascades are powerful drivers of both simple (univariate) and complex (multivariate) facets of biodiversity[28,51,53,90].

Our finding that heterogeneity in facilitation cascades enhances abundance, taxonomic richness and community dissimilarity of habitat-associated animals has implications for conservation and management. For example, consideration of the effects of secondary FS in assessments of biodiversity can lead to more effective decisions about where to commit resources to preserve and enhance biodiversity such that managers may aim to prioritise areas where high amounts of secondary FS produce spill-over effects on adjacent communities[57,91]. This study also supports the idea that morphological heterogeneity is of universal importance[7] and therefore highlights that ecosystems with many different co-occurring secondary FS should be preferentially preserved. Such systems include old forests[92], individual large trees[93], seaweed forests[94], mangroves[95] and bivalve reefs[96]. Finally, since there is a call for systematic harnessing of positive species interactions to

enhance ecosystem restoration efforts and resistance to climate change[97–99], our study highlights that facilitation cascades and the heterogeneity they generate should be taken into account when designing restoration and conservation strategies. For example, by inoculating trees in rainforest restoration projects with different sizes of different epiphyte species (characterised by different morphologies), animal biodiversity will likely increase much faster than through natural succession (as rainforest epiphytes generally have low dispersal and establishment rates)[100,101]. Not only can facilitation cascades be constructed with living organisms, but our study suggests that mimics made of artificial materials can be deployed to mirror effects of primary and secondary FS and thereby improve the heterogeneity and efficacy of constructions[34]. This approach is currently being trialled on moderately large scales during construction of new seawalls, super-trees, 3D-printed reefs[102–104], and by recycling shells from aquaculture to (re)create mussel and oyster reefs[105,106]. Thus, restoration projects based on rehabilitation of both primary and secondary FS could foster synergistic biodiversity benefits[107]. However, excessive amounts of secondary FS may inhibit facilitation cascades[60–65], in which case restoration projects could employ approaches such as culling biomass (e.g. by pruning tree epiphytes) or reducing resources, like nutrient-runoff, that fuels blooms of epiphytes and entangled alga in seagrass beds. There is also a caveat that climate-driven changes in the morphology and phenology of primary and secondary FS might, through simplification or mismatches dampen their positive effects on species diversity. Knowledge of how life-traits of primary and secondary FS will be shaped by future environmental conditions therefore may be key to sustaining efficiency of restoration plans.

We conclude that our geographically distributed experiments provide a strong mechanistic insight into the fundamental importance of habitat-associated heterogeneity in driving patterns of biodiversity in facilitation cascades. Future studies could clarify additional facets of heterogeneity that are likely to promote biodiversity in facilitation cascades, for example by experimentally manipulating interstitial refugia[108] and measure associated interstitial volume and fractals[2], compare underpinning mechanisms between landscapes (e.g. forests vs. grasslands)[15], within landscapes (e.g. tree species within forests)[11] and between individual habitat-forming organisms (e.g. trees with and without epiphytes)[this study,23], apply continuous heterogeneity treatments[33], and measure the size distributions, niche-breadth and dispersal traits of the habitat-associated animals[33]. Finally, we highlight the utility of geographically distributed factorial experiments to test for contingency of ecological processes across ecosystems and habitats, so that biodiversity models transcend a narrow basis and consider the multivariate nature of ecological communities.

## Methods

To test our hypotheses, we completed 22 factorial field experiments that compared communities associated with primary FS alone (controls) vs. together with co-occurring secondary FS, varying in amount (low vs. high) and function (dead/mimic vs. alive) to give four experimental facilitation cascade treatments. The 22 experiments represent different, yet common, facilitation cascades, many of which have been studied in detail in different ecological contexts, e.g. see[26–30,36,37,57,66,109–111].

The experiments were done in natural habitats with low levels of anthropogenic habitat-alterations and with few non-native species (although experiment 16 and 18 had as secondary FS the non-native kelp *Undaria pinnatifida*, which has colonised open coastlines throughout much of New Zealand). This contrasts other global heterogeneity studies that have focused on modified habitats, like seawalls in harbours, that can be dominated by fouling and invasive species[74]. Controls remained free of secondary FS following initial removal either because secondary FS did not colonise the primary FS (most experiments) or because they were removed throughout the experiments. The 'high' amount treatments were set at the upper end of naturally observed biomass (per unit area) of secondary FS at our field

 

sites (or an equivalent volume for dead/mimics) and the 'low' amount was set at c. 25% of the high treatment (Source Data file). If biomass of secondary FS was lost early in the experiment, treatments were re-applied. The non-living secondary FS treatments were constructed of artificial materials or, for shell-formers, by using dead (and cleaned) organisms, to mimic the structural attributes of the live secondary FS (Supplementary Table 1). Dead shells differ from their living counterparts by being 'hollow' structures (i.e. no animal or fill inside). However, facilitation effects from dead shells were similar from those from constructed mimics (unpubl. Thomsen) suggesting that dead shells and mimics had similar functional heterogeneities. Three of the experiments were conducted twice to test if our results were temporally robust (experiment 3–4, 14–15, 21–22, Fig. 1). These experiments were repeated using the same treatments, had similar experimental durations and at relatively similar locations (within 50 m, Supplementary Table 1).

The morphology of the primary and secondary FS were enumerated with a 'shape value'; 1 for simple shapes like cylinders with no or few simple interstitial spaces (e.g. kelp stipes, tree branches, mangrove prop-roots or worm tubes), 2 for intermediate shapes with medium levels of interstitial spaces (like flat forms in clusters of seagrass, saltmarsh grass or seaweed blades), and 3 for convoluted shapes with many interstitial spaces (like finely branched seaweeds or irregular oyster aggregations; see Fig. 1). Subtracting shape-values of the secondary from the primary FS showed that 1, 10, 3 and 8 experiments had 'Δmorphologies' of −1, 0, 1 and 2, respectively (Supplementary Table 1).

Each of the 22 experiments was replicated at two sites, spaced >500 m apart (see Supplementary Table 1 for geolocations), to account for the effect of random environmental variation and increase generality. Within each site, there were at least three replicates per treatment, randomly interspersed. Experiments were done in marine (e.g. kelp forest, seagrass beds, mussel reefs), transitional (e.g. intertidal mangroves and saltmarshes) and terrestrial (deciduous forest) habitats in different biogeographical regions (North America, Australasia and Europe; Fig. 1, Supplementary Table 1). Mimics of secondary FS were initially free of animals, and we therefore also removed all (>99%) animals from live FS prior to experimental manipulations through shaking, brushing, and/or washing. Experiments ran from 2 to 26 weeks (marine experiments generally being shorter as in other cross-ecosystem comparisons[8,80], matching previous experiments that have demonstrated community colonisation within this time-frame[28,31,36,37,39,53,57].

At the end of the experiments, primary and secondary FS and their associated communities were bagged together (primary FS alone for the controls) and transported to the laboratory for processing. Clonal primary FS, such as marsh grasses and seagrasses, were sampled with quadrats or cores, with sampling units smaller than the entire primary FS but larger than the secondary FS (Fig. 1). By comparison, small primary FS, such as seaweeds and molluscs, as well as mangrove pneumatophores, which were considered as a subunit of the FS, were sampled in their entirety. In the laboratory, associated animals were removed from FS by sieving through a 250 μm mesh, prior to being counted and identified to operational taxonomic units, of mixed taxonomic resolution[112]. Common, large and/or conspicuous animals were generally identified to species level while identification of small inconspicuous, rare and/or cryptic taxa was to coarser resolution (e.g. family, order, or class). To test if different resolutions used across taxonomic groups and experiments affected results, effect of heterogeneity on richness and community structure were analysed on both mixed and class levels (but not total community abundances that is unaffected by taxonomic resolution).

**Statistical analysis**. To standardise abundance and richness data we calculated Log response ratios for each individual sample with a secondary FS (at each site in each experiment). To do so, data in the control treatments within a 'site × experiment' combination were averaged and used as the denominator. Log response ratios signify the strength of facilitation cascades, with ratios larger than zero indicative that the presence of the secondary FS increases biodiversity compared to absence of a secondary FS. A few samples were not inhabited by any animals, precluding calculations of Log response ratios. We therefore did two analyses: one omitting responses with zeros (774 data points in total), and the other where all data were +1 transformed (835 data points in total, Source Data file). The Log response ratios were analysed with full linear mixed-effects models (using the R lme4 package) where (1) amount, (2) function, and (3) Δmorphology (with three complimentary tests, see below) were main test factors and the standardised biomass of the (4) primary and (5) secondary FSs, (6) absolute site latitude, (7) site longitude, (8) elevation (meters above mean sea level), (9) experimental start date, (10) experimental duration (in days) and (11) the season when the experiment was done (cold or warm), were included as fixed-effects covariates. In these analyses, factors 1–3 represented our main hypotheses (Fig. 1), 4–5 were within-experiment fine-scale resolutions of amounts, 6–8 were uncontrolled spatial moderators and 9–11 were uncontrolled temporal moderators. We also included experiment nested in site as a random effect to reduce potential non-dependence issues such as multiple effect sizes sharing the same control in one experiment. Collinearity was checked among covariates (using the R car package) and the variance inflation factor was <4 for all covariates (including the amount treatment and standardised biomass of FSs, so they were all included in the analyses). Using the R MuMIn package, we ran automated multi-model inference, including all the variables given above, and calculated model-averaged parameter estimates over the set of models with ΔAICc ≤ 2, weighting single-model estimates by their Akaike weights. We also

calculated the model-averaged importance of each covariate by summing the Akaike weights of all models in which that candidate covariate appeared. Results were consistent between the full linear mixed-effects models and the automated multi-model inference for both the full dataset and the data with zeros omitted (Supplementary Data 1). All analyses are shown in Supplementary Data 1 but we here focus on the full model (because it is simpler to compare significant and non-significant terms) and full dataset (because zero values in samples represent important ecological information about facilitation cascades).

Testing for heterogeneity effects on community structure requires multivariate analysis and therefore data-aggregation across individual samples. This analysis was therefore done as a simpler three-way fixed analysis of variance that only tested for interaction effects between amount, function and Δmorphology. For each site and experiment, Bray–Curtis multivariate dissimilarity coefficients were calculated between controls and each of the four treatments, using the PRIMER statistical package to give four Bray–Curtis values per experiment and site[76]. The random site factor was non-significant (p > 0.67) and therefore removed from the factorial design to double the replication level per experiment. A few test-factors had heterogenous variances, but data were not transformed, because no single transformation solved the problem across all test-factors, and because transformations can distort interpretations of interaction effects. In our analyses, morphological heterogeneity was assessed using qualitive rankings of the primary and secondary FS, a common approach in traditional form-functional ecology[54–56]. Facilitation cascades were therefore calculated and described as 'simple' (Δmorphology ≤ 0), 'intermediate' (Δmorphology = 1), or 'complex' (Δmorphology > 1). Three complementary sets of tests were done to analyse if this classification was robust: 11 simple vs. 8 complex Δmorphologies, 11 simple vs. 3 intermediate vs. 8 complex Δmorphologies and 11 simple vs. 11 (pooled) intermediate and complex Δmorphologies. Results were very similar between the tests, and we therefore only show the results from the latter analysis (all results are shown in Supplementary Data 1 and Supplementary Tables 2–3).

Key statistical results are presented with means and 95% confidence limits for each of the eight combinations of habitat heterogeneity (Figs. 1–2). Important significant contrasts, i.e. where confidence limits do not overlap, can therefore be identified directly from graphs. To rank the many test factors in order of relative importance we calculated the percent of Sum of Squares that was explained by each test-factor relative to all explained Sum of Squares (Supplementary 2–3). Finally, we tested for temporal consistency of heterogeneity effects, using data from the three experiments that were conducted twice. We found no effect of season on any responses (Supplementary Tables 5–6) suggesting, in concert with results from the main statistical analyses that showed low Sum of Squares related to temporal and spatial moderators (Supplementary Data 1), that our results are robust in both space and time.

**Reporting summary**. Further information on research design is available in the Nature Research Reporting Summary linked to this article.

## Data availability
All data generated or analysed during this study are included in this published article, online supplementary tables, supplementary data, and its online supplementary Source Data file. Source data are provided with this paper.

## Code availability
All R-codes are included in this published article and its supplementary information file (Supplementary Notes).

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

## Acknowledgements

Brian Mason Trust, New Zealand (M.S.T.). Australian Research Council DP170100023 (T.W.). Australian Research Council FT140100322 (P.G.). UKRI Future Leaders Fellowship MR/S032827/1 (D.A.S.). New Zealand Ministry of Business, Innovation and Employment, Contract CAWX1801: Shellfish Aquaculture (P.M.S.). New Zealand MBIE contract UOCX1704: earthquake recovery (D.R.S., M.S.T.). EU grant agreement No 869300 (F.B.).

## Author contributions

M.S.T. organised the experimental protocols and data-management. M.S.T. and Q.H. analysed the data. All authors (M.S.T., A.H.A., C.A., M.J.B., F.B., R.F., V.M.M.F., P.E.G., S.B.H., Q.H., M.K., J.L., B.S.L., L.M., Y.M., S.O., A.P.R., D.R.S., T.S., A.S., B.R.S., D.A.S., P.M.S., T.W., S.Z., G.Z.) contributed equally to undertaking field experiments and paper development.

## Competing interests

The authors declare no competing interests.

## Additional information

**Peer review information** *Nature Communications* thanks Louise Firth and the other anonymous reviewer(s) for their contribution to the peer review this work. Peer reviewer reports are available.

