## [Peer Review File · Nature Communications]

Reviewers' Comments:

Reviewer #1:

Remarks to the Author:

The manuscript "Heterogeneity within and among co-occurring foundation species increases biodiversity" presents results from a nice collaborative experiment across the globe. The focus is on founder species here. I think the experiments are well conducted and together the data set presents a nice piece of work.

However, I have a number of major concerns which should be considered in a revised version:

- Many abbreviations are used in the manuscript, which make it very difficult to read. I think you can do without them.
- I lack a clear conceptual derivation of the heterogeneity axes. Habitat amount, for example, is not part of heterogeneity per se. According to the more energy hypothesis, more species live in a large habitat by pure chance only because of the higher population sizes. This is addressed in the text but is not clearly separated and confuses the reader. I recommend a clear revision of the conceptual part.
- Species richness is a complicated measure. As far as I understand the description, it is rather species density (see Gotelli & Colwell 2001 *Ecol Letters*).
- The mixing of different taxonomic resolutions (Species, Family, Order) is also difficult here. Since adaptation to heterogeneity happens at the species level, mixing is questionable here.
- The graphical representation of the main results is very cryptic. Here, a form of presentation would be desirable that makes it easy for the reader to recognise independent main effects and additive or shared effects.

Minor comments:

- Line 45: Environmental and habitat heterogeneity: environment is part of the habitat?
- Line 48: this is only one obstacle, there is a clear theory and also the area trade off effect!
- Line 58-59: This sentence is too weak. Make a clear implication!
- Line 68-69: this is misleading because MacArthur created the foliage height diversity it is not the biomass of trees or the height itself
- Line 106/107: this is not due to heterogeneity
- Line 142-150: the significance is not surprising in such a large data set. the more interesting is the relative explained variance of the three axis.
- Line 334: why not as offset?
- Line 367: only three makes no sense I would include them all in one model and remove this aspect, it weakens the approach.

Reviewer #2:

Remarks to the Author:

The paper by Thomsen et al. investigated by means of globally distributed factorial field-based experiments how habitat-associated environmental heterogeneity influences biodiversity. Based on their analyses covering multiple primary foundation species (pFS), the authors claim that the habitat (amount, function and morphology) had positive and additive effects on biodiversity. A combination of increasing habitat area by secondary (living) foundation species (sFS) (beyond that provided by pFS), additionally protective microhabitats and other resources (e.g. food) were identified as strong driving forces catalyzing functional cascades (FCs) and resulting in the enhancement of biodiversity (measured as abundance, species richness and Bray Curtis dissimilarity). In particular, the authors showed that morphological differences between secondary and primary foundation species influence FCs and that even though not contributing with resources, mimics of FS can still contribute to the enhancement of biodiversity.

The claims are not novel and have been recurrently presented in different papers namely regarding the role of pFS, as the authors explain in their introduction. But here the authors provide evidence at the global scale based on concerted collections (not fully standardized) and combining a range of pFS. In this sense, I find the paper comprehensive and of interest to the scientific community as rather than compiling results already published coordinated experiments are conducted increasing the significance of the conclusions.

Overall, I find the study (and statistical analysis) robust but I would appreciate some additional clarification on the methods. As it is, I do not think it is possible to reproduce their work. In some cases, I believe it is enough to shuffle some sentences around but in others (see below), information is in my opinion missing.

Specific comments

Methods

Line 282: Based on what were the sFS established? Depending on the target animals, the size of secondary FS can change. For example, for an amphipod, small epiphytes associated with seagrass leaves might be relevant. I would like to better understand the criteria used to classify the sFS tested.

Line 282: clarification about low and high (amount) is needed. These terms are too vague. They are explained later (lines 304-307) but I would consider moving up those lines.

Line 289: You make reference to a reclassification of the 22 experiments to reach a balanced design. Although I understand the importance of reaching a balanced design, I would like to know what such reclassification represent. Please, clarify.

Line 295: two sites do not allow you to "ensure that conclusions about FCs were locally representative". Please, tone it down.

Line 303: add "from" before those

Lines 304-307: please explain how the quantifications were done. If you want, detailed information about the methods can be provided in SM, but you talk about natural gradients in biomass without providing any further clarification.

Line 309: what was the reasoning behind such a large range in the duration of the experiments? If this is related to specific habitats, information should be provided. Was the period consistent when the same pFS was being tested?

Information about the controls is needed. How have you assured that sFS were not colonizing and biasing the results?

Lines 367-370: Please provide more information regarding the repeated experiments. was it a proper seasonal analysis or have you looked at temporal variability? was the extent of the experiment maintained?

Line 203: discussed instead of discusses

Line 248: One of the issues I expressed in the methodology is related to the concept of low and high but the authors have provided some explanation about it. But when it comes to management and conservation, these terms cannot be applied as they will be meaningless to practitioners.

Moreover, you also mention in the discussion that the amount of sFS above certain points can create adverse environmental conditions to the pFS and, consequently, to the ecosystem. I think this should be carefully addressed and if recommendations for management and conservation are provided that they should be better contextualized.

Reviewer #3:

Remarks to the Author:

I commend the authors on this research. This paper is very well written and executed and I congratulate them for conducting global experiments across environments. Through conducting 22 experiments across terrestrial and marine environments the authors have found strong evidence to support the importance of habitat heterogeneity in promoting biodiversity via cascades of facilitative interactions. Whilst this is perhaps not surprising, it has not been demonstrated empirically across systems in this fashion. I recommend this paper for publication.

The results from this study will be of interest across a broad spectrum of fields and systems.

There are lots of questions that can be asked about the details of the FS used, for instance, how do you account for the condition/quality or age of the pFS and sFS used in the experiments. But I appreciate that across systems this level of detail is impossible to provide. I know the authors state that 6 months is a typical duration for running such experiments, but arguably 6 months is very short and indeed there will undoubtedly be a seasonal component to this. It would be good to see some mention of the timing/seasonality aspect in the methods.

What is the justification for the focus on animals? On line 94 and 107, it refers to animals, but equally plants/algae can also benefit from FCs.

I was expecting to see some mention of the stress gradient hyp. Results would be expected to vary under different environmental conditions and under different community assemblies... for instance, in highly disturbed environments the local community might be dominated by invasive species which may be more or less likely to colonise and or/benefit from the pFS or the sFS. The application of facilitation theory and facilitation cascade theory to management of disturbed, urban and novel ecosystems may be misguided. Were all of your experiments carried out in natural/relatively unaltered habitats. It would be good to at least mention this in the methods, and acknowledge this potential limitation (if it is the case). For instance, a recent global analysis (Strain et al. 2021, Global Ecology and Biogeography) of the seeding of bivalves (pFS) onto seawalls yielded conflicting results, highlighting that responses to heterogeneity are context dependent, and may differ in highly disturbed environments.

More from curiosity perspective, I wondered were there any differences in responses if you separated animal and plant/algal pFS and sFS? I am not suggesting that you do this, like I say, just curious. This doesn't require a response.

Were any of your FS invasive in the local context? As invasives are less likely to be colonised (enemy release), it is worth clarifying here.

L77-81 - Its not clear what you mean by habitat function here and how this links to biodiversity (L78). It's a bit of a chicken and egg. The habitat with greater numbers of taxa support more functions. The examples given about trees/woodpeckers and mistletoe do not help to elucidate this. Suggest developing this argument better here.

L78 – better to say resource use rather than resource extractions?

L99 – ecosystem should be singular

L132 – and presumably functions too?

L293 – experiments should be plural

L355 – non significant rather than insignificant

L386 and 387 - should Crassostrea now be Magallana

Reviewer #1 (Remarks to the Author):

The manuscript “Heterogeneity within and among co-occurring foundation species increases biodiversity” **presents results from a nice collaborative experiment across the globe.** The focus is on founder species here. **I think the experiments are well conducted and together the data set presents a nice piece of work.** However, I have a number of concerns which should be considered in a revised version:

- Many abbreviations are used in the manuscript, which make it very difficult to read. I think you can do without them.

- ***Done. We have removed all abbreviations minus ‘FS’ (for ‘foundation species’) that is used to shorten the text. We describe this single abbreviation the first time it is used under each heading.***

- I lack a clear conceptual derivation of the heterogeneity axes. Habitat amount, for example, is not part of heterogeneity per se. According to the more energy hypothesis, more species live in a large habitat by pure chance only because of the higher population sizes. This is addressed in the text but is not clearly separated and confuses the reader. I recommend a clear revision of the conceptual part.

- ***Done. We have clarified the concept underlying heterogeneity axes. For example, we have added a schematic figure that contrasts the three axes of heterogeneity (Fig 1C) and we have also improved the data graph (Fig 2) to better contrast the three types of heterogeneity.***
- ***However, we contend that ‘different amounts’ is indeed a form of ‘heterogeneity’ and have therefore not changed this specific terminology. For example...***
 - ***Wikipedia (accessed 2/5-21) states that “Homogeneity and heterogeneity are concepts often used in the sciences and statistics relating to the uniformity of a substance or organism. A material or image that is homogeneous is uniform in composition or character (i.e., colour, shape, size, weight, height, distribution, texture, temperature, architectural design, etc.); one that is heterogeneous is distinctly nonuniform in one of these qualities”. Clearly, size, weight and height are all attributes related to amounts.***
 - ***Webster’s Revised Unabridged Dictionary describes heterogeneity as ‘Differing in kind; having unlike qualities; possessed of different characteristics; dissimilar’ – i.e., *having different amounts also qualifies as heterogeneity.****
 - ***Cambridge dictionaries say heterogeneity is ‘consisting of parts or things that are very different from each other:’ Again, *small vs. large amounts are different from each other and therefore ‘heterogeneous things’.****
 - ***Importantly, the seminal meta-analysis by Stein et al¹ that correlated environmental heterogeneity and taxonomic richness included several habitat-amount variables in their comprehensive analyses (e.g., % cover of forests, plant density, tree trunk perimeters, patch sizes).***
 - ***Furthermore, Tokeshi & Arakaki’s insightful review paper (that used heterogeneity and complexity as synonyms)² about relationships between biodiversity and habitat heterogeneity included both sizes and abundances of ‘ecological elements’ (where elements in our case study correspond to foundation species) in their list of essential attributes.***
 - ***Finally, the seminal paper by McArthur³ about habitat heterogeneity included data related to both canopy height and canopy width – attributes that are related to habitat-amount (the higher and wider the canopies – the higher the amount of habitat) in a Foliage Height Diversity index.***

•Species richness is a complicated measure. As far as I understand the description, it is rather species density (see Gotelli & Colwell 2001 Ecol Letters).

- ***Done. Yes, the reviewer is correct. We have added 'species density' (in brackets) and the Gotelli & Colwell reference after the first time we use the term 'richness'. However, we continue to refer to 'richness' throughout the manuscript because, although Gotelli & Colwell make an important technical point about this terminology, 'richness' is still the default terminology used to describe the number of taxonomic units in a sample, e.g. ^{4 5,6 7-12}. In other words, we do not want to confuse the reader with a species-density-terminology that has not become mainstream and generally accepted.***

•The mixing of different taxonomic resolutions (Species, Family, Order) is also difficult here. Since adaptation to heterogeneity happens at the species level, mixing is questionable here.

Done. We thank the reviewer for pointing out the importance of taxonomic resolution and now address this issue with additional statistical analysis. Please note the following:

- 1. This critique does not affect our abundance results that remain the same irrespective of taxonomic resolution.***
- 2. We added new analysis based on assigning all individuals to the class level. Unfortunately, we could not repeat the analysis at the level of Orders because some animals, like juvenile copepods, seastars, and foraminifera, could not be identified to this level. The class-level analysis showed very similar results to the analysis done on mixed/operational taxonomic units - with strong positive effects of facilitation cascades. This result highlights that our three axes of heterogeneity are of fundamental importance in controlling biodiversity across scales, systems, and taxonomic units.***
- 3. We disagree that adaptations to heterogeneity only happen at the species level. Instead, we argue that adaptations can operate across all taxonomic resolutions. For example, virtually all small marine mobile organisms benefit from small interstitial spacing (e.g., to avoid predation or intertidal desiccation stress). In this case it is organismal size – not phylogeny per se – that selects for the strength of facilitation (which is also why early life stages of organisms that are large as adults – e.g., microscopic juvenile urchins and seastars are often independently recorded in facilitation cascade studies). Similarly, most mobile invertebrate grazers consume the same type of plant or algae material, irrespective of their phylogeny (e.g., grazing isopods, limpets, and urchins respond to live vs. mimic foundation species in similar ways) ¹³⁻¹⁷. In short, heterogeneity transcends many types of species-levels adaptations – hence the arise of a 'functional ecology' – and a reason why it is common to use mixed taxonomic levels in marine studies ¹⁸⁻²⁴ [note also that operational taxonomic units (OTUs) traditionally was used to represent mixed taxonomic level analysis – but the rise in genetic techniques implies that today, OTU is more commonly used in molecular studies – however, in these cases we – like many other – prefer the more precise MOTU terminology (molecular operational units) ²⁵⁻²⁷.***
- 4. The Linnean classifications system is a human-made abstraction. Indeed, a class for one type of organism may better correspond to a subphylum for another type of organism (and similar arguments can be made for lower levels). This problem is particularly relevant for many of the animals we quantified – with large numbers of individuals being less than 1 mm long and poorly***

described in the scientific literature. The advantages of using mixed taxonomic units in our study (that transcended ecosystems and habitats) are that mixed taxonomic units can cut through much of this resolution noise without having to analyse all communities with advanced genetic methods.

- 5. Our results are most likely conservative, and we would probably have found even stronger results on richness and Bray-Curtis dissimilarity if all individuals could have been identified to species (e.g., using state-of-art genetic methods and consulting the best taxonomy specialist around the world). However, as stated, we could not identify some individuals to species (e.g., nematodes, foraminifers, small copepods, or ostracods or microscopic juvenile seastars).*

•The graphical representation of the main results is very cryptic. Here, a form of presentation would be desirable that makes it easy for the reader to recognise independent main effects and additive or shared effects.

- Revised as suggested. We have changed the data graphs (Fig. 2) to better reflect the 3-factorial approach – changing and splitting the bar graphs into multiple paired mean-plots and using the same distinct symbols, sizes, and colours to represent the three crossed test-factors that we used in Fig. 1 to explain our methods. We have also replaced the SE with 95%CL, so it is easier for readers to identify significant contrasts within and between plots. In other words – if 95%CL do not overlap, the reader knows that the means are different. Finally, we have added more information to the legend to make the figures easier to understand as stand-alone information.*

Minor comments:

•Line 45: Environmental and habitat heterogeneity: environment is part of the habitat?

- Done. In this sentence we have kept the word ‘environment’ (broad) but have removed the ‘habitat’. However we argue that environment is the broader more general term – that is, habitat is nested within the environment (e.g. as in ¹)*

•Line 48: this is only one obstacle, there is a clear theory and also the area trade off effect!

- There are many reasons for why we do not fully understand the universal role of heterogeneity in driving biodiversity. However, in a compact abstract we do not have space to discuss these many theories – we simply highlighted that one explanation can be the lack of coordinated experiments across scales and habitat types. Our purpose was to set up the reasoning for conducting the study within the brief confines of an abstract. We provide a more nuanced explanation of heterogeneity-biodiversity relationships in the introduction.*

•Line 68-69: this is misleading because MacArthur created the foliage height diversity it is not the biomass of trees or the height itself

- Done. This is of course correct, and we thank the reviewer for pointing it out – MacArthur did not measure biomass but used simpler non-destructive proxies (which included width and height of strata – combined into an index as discussed in our prior response to Reviewer 1 above). We have corrected the text to reflect this, but the take-home message is the same – width, number and*

height co-vary with biomass/amount. That is, the wider and longer the canopies are likely associated with higher amounts of canopy habitat and are reflected in higher the bird diversity.

•Line 106/107: this is not due to heterogeneity

- ***We disagree; 'different amounts' is a form of 'heterogeneity' - See our previous detailed comment that highlighted several standard heterogeneity definitions from dictionaries and key research papers that included 'different amounts' as heterogeneity (where amount is an umbrella term for biomass, volume, height, width, length, density, and size).***

•Line 142-150: the significance is not surprising in such a large data set. the more interesting is the relative explained variance of the three axis.

- ***Done. We agree - which is why we reported sum of square values after the p-values. To highlight this important take-home message, we have added more sentences to the methods, results and discussion about sum of squares and relative importance of different test-factors in ANOVA²⁸. This analysis highlights (a) that our heterogeneity test are ecologically important across test factors, systems, responses and taxonomic resolutions, (b) that the ecological function (as an independent factor) and the amount x complexity interaction generally explains most of the univariate data-variability, (c) that delta-morphology explains most of the variability for the multivariate analysis and (d) that the 3 spatial and 3 temporal non-controlled covariates explain far less data variability, thereby highlighting the particular importance of heterogeneity across spatiotemporal scales.***

•Line 334: why not as offset?

- ***We do not understand what the reviewer means about 'off-set' – this section is about how we collected primary and secondary foundation species.***

•Line 367: only three makes no sense I would include them all in one model and remove this aspect, it weakens the approach.

- ***Done. We have added the suggested analyses (11 simple vs. 8 most complex delta-morphology) – the results from these new tests (added to the online supplements) were very similar to our other test results so we still focus on the 11 simple vs. 11 complex test-comparison in the main manuscript.***

Reviewer #2 (Remarks to the Author):

The paper by Thomsen et al. investigated by means of globally distributed factorial field-based experiments how habitat-associated environmental heterogeneity influences biodiversity. Based on their analyses covering multiple primary foundation species (pFS), the authors claim that the habitat (amount, function and morphology) had positive and additive effects on biodiversity. A combination of increasing habitat area by secondary (living) foundation species (sFS) (beyond that provided by pFS), additionally protective microhabitats and other resources (e.g. food) were identified as strong driving forces catalyzing functional cascades (FCs) and resulting in the enhancement of biodiversity (measured as abundance, species richness and Bray Curtis dissimilarity).

- ***Please note that FC was an abbreviation for Facilitation Cascades – not Functional Cascades. This confusion is now impossible in the revised manuscript because we removed the abbreviation, i.e. we now write ‘facilitation cascades’ instead of FC.***

In particular, the authors showed that morphological differences between secondary and primary foundation species influence FCs and that even though not contributing with resources, mimics of FS can still contribute to the enhancement of biodiversity. The claims are not novel and have been recurrently presented in different papers namely regarding the role of pFS, as the authors explain in their introduction. But here the authors provide evidence at the global scale based on concerted collections (not fully standardized) and combining a range of pFS. In this sense, I find the paper comprehensive and of interest to the scientific community as rather than compiling results already published coordinated experiments are conducted increasing the significance of the conclusions.

- ***We are aware that our hypotheses (‘claims’) may not be considered remarkably novel in contextual isolation and within an idiosyncratic case-study, such as within a particular habitat, geographical location or for a model-species. However, novelty arises because the 3 axes of heterogeneity are tested based on data collected...***
 - *for facilitation cascades – our research focus is on the ecological importance of secondary foundation species (the vast majority of published studies have tested for ecological importance of primary foundation species),*
 - *across bioregions, habitats, elevation levels, model species, seasons, experimental durations, and taxonomic resolutions,*
 - *in a fully crossed factorial approach to test for relative importance and interaction effects,*
 - *with controlled manipulative experiments.*
- ***In other words, it is the combination of these four points that makes the study unique and novel and allows for a new level of data interpretation and new insights. Please also note that traditional meta-analyses of published literature, of which we have done many, cannot replace the last two points (i.e., that our work presents results from fully-crossed manipulative experiments).***

Overall, I find the study (and statistical analysis) robust but I would appreciate some additional clarification on the methods. As it is, I do not think it is possible to reproduce their work. In some cases, I believe it is enough to shuffle some sentences around but in others (see below), information is in my opinion missing.

- ***Done. We added data to the online supplements and believe each experiment can now be reproduced (we added less information to the methods section the main body text of the manuscript because much of the new information is specific to individual experiments, so adding all the new information to the manuscript itself would increase its length significantly. Note that the methods described in the manuscript were initially emailed to all co-authors as the single***

instructions on how to do each experiment (and it worked). With the added information to the online supplement (including biomass data for foundation species) we believe the experiments can now be replicated.

Specific comments

Methods

Line 282: Based on what were the sFS established? Depending on the target animals, the size of secondary FS can change. For example, for an amphipod, small epiphytes associated with seagrass leaves might be relevant. I would like to better understand the criteria used to classify the sFS tested.

- ***Done. Most of the facilitation cascade experiments were designed based on our previous expert knowledge, because we have worked with these systems for many years – for many of the experiments we have described impacts from co-occurring FS on biodiversity in different contexts with mensurative and different experimental approaches e.g.,²⁹⁻⁴⁰. Given that facilitation cascade theory is an emerging research field we established an international working group to combine our knowledge from the different systems and explore ecological generalities using meta-analysis⁴¹ and experiments (this study). We have added more references to the paper so that readers can consult these for more information about most of the facilitation cascades that are experimentally manipulated here. However, the 22 experiments also include a few ‘new’ facilitation cascades where the ‘background’ mensurative data have not yet been published (this is work in progress by different co-authors).***
- ***Most importantly, the ecological relevance of each individual facilitation cascade is not the target of the present study that instead aimed to identify fundamental underpinning mechanisms that control biodiversity. A future study will instead target the ecological relevance of published facilitation cascade experiments (in other words, some facilitation cascades are – just like trophic cascades - more ecologically important and common across space and time than others).***

Line 282: clarification about low and high (amount) is needed. These terms are too vague. They are explained later (lines 304-307) but I would consider moving up those lines.

- ***Done – we have moved the explanation up as suggested.***

Line 289: You make reference to a reclassification of the 22 experiments to reach a balanced design. Although I understand the importance of reaching a balanced design, I would like to know what such reclassification represent. Please, clarify.

- ***Done. We now explain in more detail how and why we grouped the 22 experiments into morphological heterogeneities and now perform 3 supplementary sets of statistical analyses to address this issue (Line 434-442). Importantly, all analyses showed the same results highlighting that impact of morphological heterogeneity is robust across our classifications. In other words, we found the same results both when grouping morphologies into 2 and 3 levels of heterogeneity and both when we included or excluded the middle/in-between level of heterogeneity.***

Line 295: two sites do not allow you to “ensure that conclusions about FCs were locally representative”. Please, tone it down.

- ***Done – we have revised the language to tone it down. We agree that was too ‘unconditional’ a statement. Still, replication of study sites reduced impacts of local environmental noise and makes results more general (in other words, results are more general if they can be documented at two, compared to one, sites).***

Line 303: add “from” before those

- ***Done.***

Lines 304-307: please explain how the quantifications were done. If you want, detailed information about the methods can be provided in SM, but you talk about natural gradients in biomass without providing any further clarification.

- ***Done, we have added information to the online supplement about the biomass of the foundation species for each individual effect size. We have also added biomass of both primary and secondary foundation species (standardized per experiment) as co-variates to our data analysis.***

Line 309: what was the reasoning behind such a large range in the duration of the experiments? If this is related to specific habitats, information should be provided. Was the period consistent when the same pFS was being tested?

- ***Done, we have added more information and relevant references. Yes – the differences in duration co-vary with the systems being studied – forest/mangrove/marsh experiments were typically longer than fully marine studies (as often noted in meta-analyses e.g.,^{41,42}). The short incubation time in marine systems reflects that small marine taxa colonize new out-transplanted habitats very fast (i.e. within a few days)^{34,43-49} whereas colonization times is typically slower in terrestrial and marsh/mangrove systems. Furthermore, the few facilitation cascade experiments we are aware of that included repeated time series found relatively consistent strength of facilitation cascades^{34,48} suggesting that these facilitation cascades occur across variable time scales. To address the issue, we included experimental duration as a covariate in our analysis and with little on effect sizes (but we agree that experimental duration is an interesting perspective that should be included as a crossed treatment in future global experiments).***

Information about the controls is needed. How have you assured that sFS were not colonizing and biasing the results?

- ***Done – we have added a sentence to highlight this was not a problem (line 346-348). Indeed, we always recorded the amount of sFS that colonized the controls – this only happened to a few controls in a few experiments and only in extremely small amounts. Furthermore, if colonization was observed midway through the experiment the secondary FS was removed (we are aware that it can become a problem in some long-term experiments – in these cases continued maintenance and removals are required to maintain proper controls without secondary FS).***

Lines 367-370: Please provide more information regarding the repeated experiments. was it a proper seasonal analysis or have you looked at temporal variability? was the extent of the experiment maintained?

- ***Done. We have added information about the three experiments that were completed in two seasons – using similar treatments, durations, and relatively similar locations (within 10-50 m). Note that the results from these experiments are statistically independent because (a) new biomass of foundation species and new mimics were used for each experiment, (b) treatments were allocated at random and (c) spatial location of primary and secondary FS were slightly different to the previous experiment. In other words, the data are not ‘repeated measurements’. Redoing three of the experiments allowed us to do a preliminary analysis of the importance of season (it was not important).***
- ***In addition, we have added three temporal co-variables to our main analysis on the univariate responses – start date, season, and experimental duration (in addition to three spatial co-variables: latitude, longitude, and elevation). We found few and weak effects overall from both the 3 temporal and 3 spatial co-variables (all results are in the supplements - except that the elevation factor explained a relatively high percentage of the Sum of Squares in a single test (out of 12 tests). Importantly, in our main tests the 6 spatiotemporal co-variables explained <7% of Sum of Squares compared to the 3 axes of heterogeneity (i.e., >93% for amount, function, and delta-morphology).***
- ***We also note that several facilitation cascade studies have shown temporally consistent positive effects from secondary foundation species^{34,48}.***
- ***However, we agree that it would be interesting in future studies to address temporal variability as a more explicit and rigorous factorial test-factor (e.g., by redoing experiments with low, medium, and long durations in several seasons).***

Line 203: discussed instead of discusses

- ***Done.***

Line 248: One of the issues I expressed in the methodology is related to the concept of low and high but the authors have provided some explanation about it. But when it comes to management and conservation, these terms cannot be applied as they will be meaningless to practitioners. Moreover, you also mention in the discussion that the amount of sFS above certain points can create adverse environmental conditions to the pFS and, consequently, to the ecosystem. I think this should be carefully addressed and if recommendations for management and conservation are provided that they should be better contextualized.

- ***Done. We have added a few lines to the discussion about how the amounts of secondary FS can be incorporated into management and conservation and how extreme amounts may have negative impacts on animal diversity and therefore also require management (but note that exact values of whereby sFS negatively affect pFS varies widely between ecosystems - e.g., impact of drift algae in estuaries on seagrass is more severe than impact of rooted alga in open water systems⁵⁰) and environmental conditions such as when negative impact of drift seaweed on seagrass increases dramatically with temperature⁵¹.***

Reviewer #3 (Remarks to the Author):

I commend the authors on this research. This paper is very well written and executed and I congratulate them for conducting global experiments across environments. Through conducting 22 experiments across terrestrial and marine environments the authors have found strong evidence to support the importance of habitat heterogeneity in promoting biodiversity via cascades of facilitative interactions. Whilst this is perhaps not surprising, it has not been demonstrated empirically across systems in this fashion. **I recommend this paper for publication.**

The results from this study will be of interest across a broad spectrum of fields and systems. There are lots of questions that can be asked about the details of the FS used, for instance, how do you account for the condition/quality or age of the pFS and sFS used in the experiments. But I appreciate that across systems this level of detail is impossible to provide. I know the authors state that 6 months is a typical duration for running such experiments, but arguably 6 months is very short and indeed there will undoubtedly be a seasonal component to this. It would be good to see some mention of the timing/seasonality aspect in the methods.

- ***Done. We have added details to the methods and online supplements – listing seasonality, starting date and experimental duration. These three temporal characteristics have also been added as co-variables to our main data analysis. More specifically, the experimental durations are typical for these types of animal-habitat-colonization experiments - where duration often is a bit longer in forest/mangrove/marsh systems compared to marine studies (as also noted in several meta-analyses e.g.,^{41,42}). The short duration in the marine experiments reflects that small (and short lived) marine invertebrates colonize new out-transplanted habitats very rapidly (within a few days)^{34,43-49}. Furthermore, facilitation cascade experiments that have included repeated measurements have shown relatively consistent strengths of facilitation cascades over time^{34,48} suggesting that facilitation cascades occur across variable time scales. We do however also agree that duration and seasonality are important test factors to include as specific crossed treatments in future global experiments.***

What is the justification for the focus on animals? On line 94 and 107, it refers to animals, but equally plants/algae can also benefit from FCs.

- ***We agree that primary producers can also benefit from facilitation cascades (e.g., see our work on 'long' facilitation cascades⁵²). However, the relative importance of plants as 'inhabitants' (focal species/end-users) is likely to vary between ecosystems and habitats and could therefore introduce bias (this is another potential interesting future research area). Importantly, almost all facilitation cascade studies to date - and in particular short-term colonization experiments - have focused on animal responses⁴¹. We therefore also focused on animals to allow direct comparisons to past research. (As a side note, we recorded colonization by only two primary producers (Ulva and Gelidium) in a single experiment and in low abundances – these seaweed responses were not included in our analysis).***

I was expecting to see some mention of the stress gradient hyp. Results would be expected to vary under different environmental conditions and under different community assemblies... for instance, in highly disturbed environments the local community might be dominated by invasive species which may be more or less likely to colonise and or/benefit from the pFS or the sFS. Were all of your experiments carried out in natural/relatively unaltered habitats. It would be good to at least mention this in the methods and acknowledge this potential limitation (if it is the case). For instance, a recent global analysis (Strain et al.

2021, Global Ecology and Biogeography) of the seeding of bivalves (pFS) onto seawalls yielded conflicting results, highlighting that responses to heterogeneity are context dependent, and may differ in highly disturbed environments.

- ***Done. We have added information to our methods section to highlight that our experiments were done in natural and relatively undisturbed systems – forests, estuarine mudflats, open reefs, mangroves, etc. None of our experiments were done in highly modified harbours or on seawalls/piers like in the Strain et al study (these harbour/jetty systems are ecologically novel and dominated by opportunistic and small turf and filamentous algae and sessile invertebrates – in other words, simplistic ‘fouling’ communities that can grow ‘anywhere’ – including smooth ship hulls and flat seawalls).***
- ***We are aware of the potential importance of the SGH for facilitation cascade theory⁵³ and we often discuss SGH in our more specific case studies as well as reviews of facilitation cascade theory (e.g.^{41,54}). We aim to tackle this hypothesis head-on in future facilitation cascade work, but it was beyond the scope of the present study. Nevertheless, we have added a sentence stating that the SGH is an important future research topic in facilitation cascade studies (line 226-230).***
- ***We are aware that ecology is fundamentally context dependent⁵⁵ and we often discuss this in our facilitation cascade research^{30,33,34,39,41,54,56}. Indeed, the main idea about the current manuscript is to challenge context-dependency⁵⁵ and search for the elusive general rules by working across ecosystem and habitat types. Indeed, our meta-analysis of past publications⁴¹ and the present submitted work suggest that some, albeit simple, rules provide a starting point toward identifying general rules in facilitation-cascade theory.***

More from curiosity perspective, I wondered were there any differences in responses if you separated animal and plant/algal pFS and sFS? I am not suggesting that you do this, like I say, just curious. This doesn't require a response.

- ***We have explored this issue (unpublished data exploration), but we wait for future more specific studies to target this idea with rigorous quantitative analyses. The main reason we did not address plant-animal FS dichotomies here is because this research question is much more relevant to marine than terrestrial systems. In marine systems sessile animals, angiosperms and seaweed all provide analogue habitat-structures – whereas in terrestrial systems it's mainly just ‘plants’. Another dichotomy relates to sessile vs mobile foundation species⁵⁷, but mobile FS require yet another very different discussion and we don't want to confuse readers with it here. For an example of how we address this plant-animal and sessile-mobile dichotomies in invasion biology see⁵⁸. To keep the present paper as general as possible and not detour into discussion about reconciling with terrestrial studies that typically do not equate animals with ‘habitat-forming foundation species’), we decided to address this topic in future studies aimed specifically at this question.***

Were any of your FS invasive in the local context? As invasives are less likely to be colonised (enemy release), it is worth clarifying here.

- ***We have very few invasive species in our data set – both as foundation species and animal responses. The exception is the sub-experiment with Undaria as secondary FS - a non-native kelp that today inhabits many open coastlines around the world – see our recent reviews for details^{59,60}. We have added that Undaria is a non-native species to the method section, but that the experiment was done on healthy and diverse open coast reefs dominated by native species.***

Importantly, Undaria is equally likely to be colonized by small invertebrates as native kelps (work in progress) and we have previously documented that Undaria holdfasts are inhabited by invertebrate assemblages that are relatively similar to native seaweeds³⁷⁻³⁹. Indeed, 'tight coevolution' does not appear to be a strong predictor of the strength of relationship between foundation species and their associated animals in marine systems as evidence from studies with invasive foundation species (e.g., see our old work on epibiota associated with non-native Sargassum vs. native Halidrys and various reviews about epibiota and invasive marine species)^{20,61}. This low level of host-specificity was demonstrated in our study where we found animals easily colonized abiotic mimics.

L77-81 - Its not clear what you mean by habitat function here and how this links to biodiversity (L78). It's a bit of a chicken and egg. The habitat with greater numbers of taxa support more functions. The examples given about trees/woodpeckers and mistletoe do not help to elucidate this. Suggest developing this argument better here.

- *Done. We have reworded this text about habitat function, heterogeneity, and links to biodiversity (line 79-88). More generally, we resolved this chicken-and-egg quandary because we experimentally separate the effect of the secondary foundation species (that is fixed, controlled, manipulated = either 0 or 1 species) from the animals that depend on it (i.e., the uncontrolled response variable, that can be any number of colonizing animals). It is therefore valid to hypothesize that a secondary FS (of which there can only be 0 or 1 species) that provides >>1 function (many types of resources) will support higher animal-biodiversity because it allows species with different traits and niches to use different types of resources, compared to a secondary FS that only provides a single function (e.g., only space).*
- *The woodpecker represented a widely recognized classic example from the functional landscape/habitat literature whereas the mistletoe example is well known from the facilitation cascade literature⁶²⁻⁶⁸.*
 - *For example, a hypothetical forest could be dominated by 100 softwood tree species that are free of parasite/mistletoe (i.e., this forest is composed of 100 different FS), but this diverse forest would be without woodpeckers or mistletoe birds. By replacing only 2 of the soft tree species with different FS – such as a hardwood tree species (e.g., Quercus pyrenaica) and a mistletoe parasite – the richness of FS would remain constant (100), but the function of the forest would increase dramatically because vital nest and food resources would now be available for woodpeckers and mistletoe-birds. In other words, this functional approach emphasizes that FS provide more than just a space to live in and on, and that the traits and resource requirements of the animals that depend on FS are important.*
- *We tested this functional approach by hypothesizing that living FS provide more functions than FS that are morphologically similar but not alive. Specifically, living FS provide at least 5 types of functions/resources (of which non-living mimics of FS only provide the first 2): (1) habitat for attachment, resting or feeding and (2) space to hide from predators and avoid abiotic stress (e.g. from intertidal desiccation or from waves), (3) a direct food resources because animals can browse on the living FS, (4) waste products that can support decomposer species, and (5) altered biogeochemistry through its metabolic processes that may increase oxygen levels, water flow or provide biogeochemical microhabitats.*

L78 – better to say resource use rather than resource extractions?

- ***Done. We now say acquire resources instead of extracting resources.***

L99 – ecosystem should be singular.

- ***Done.***

L132 – and presumably functions too?

- ***Done. Yes – this is now hypothesis 2.***

L293 – experiments should be plural

- ***Done.***

L355 – non significant rather than insignificant

- ***Done.***

L386 and 387 - should Crassostrea now be Magallana

- ***Done.***

References used in replies

- 1 Stein, A., Gerstner, K. & Kreft, H. Environmental heterogeneity as a universal driver of species richness across taxa, biomes and spatial scales. *Ecology letters* **17**, 866-880 (2014).
- 2 Tokeshi, M. & Arakaki, S. Habitat complexity in aquatic systems: fractals and beyond. *Hydrobiologia* **685**, 27-47 (2012).
- 3 MacArthur, R. H. & MacArthur, J. W. On bird species diversity. *Ecology* **42**, 594-598 (1961).
- 4 Grace, J. B. *et al.* Integrative modelling reveals mechanisms linking productivity and plant species richness. *Nature* **529**, 390-393 (2016).
- 5 Bell, T., Newman, J. A., Silverman, B. W., Turner, S. L. & Lilley, A. K. The contribution of species richness and composition to bacterial services. *Nature* **436**, 1157-1160 (2005).
- 6 Orme, C. D. L. *et al.* Global hotspots of species richness are not congruent with endemism or threat. *Nature* **436**, 1016-1019 (2005).
- 7 Nogués-Bravo, D., Araújo, M., Romdal, T. & Rahbek, C. Scale effects and human impact on the elevational species richness gradients. *Nature* **453**, 216-219 (2008).
- 8 Downing, A. L. & Leibold, M. A. Ecosystem consequences of species richness and composition in pond food webs. *Nature* **416**, 837-841 (2002).
- 9 Adler, P. B. *et al.* Productivity is a poor predictor of plant species richness. *science* **333**, 1750-1753 (2011).
- 10 Damschen, E. I., Haddad, N. M., Orrock, J. L., Tewksbury, J. J. & Levey, D. J. Corridors increase plant species richness at large scales. *Science* **313**, 1284-1286 (2006).
- 11 Grace, J. B. *et al.* Response to comments on "Productivity is a poor predictor of plant species richness". *science* **335**, 1441-1441 (2012).
- 12 Fraser, L. H. *et al.* Worldwide evidence of a unimodal relationship between productivity and plant species richness. *Science* **349**, 302-305 (2015).
- 13 Thomsen, M. S. *et al.* Habitat cascades: The conceptual context and global relevance of facilitation cascades via habitat formation and modification. *Integrative and Comparative Biology* **50**, 158-175 (2010).
- 14 Bologna, P. A. & Heck, K. L. Macrofaunal associations with seagrass epiphytes - Relative importance of trophic and structural characteristics. *Journal of Experimental Marine Biology and Ecology* **242**, 21-39 (1999).
- 15 Gartner, A., Tuya, F., Lavery, P. S. & McMahon, K. Habitat preferences of macroinvertebrate fauna among seagrasses with varying structural forms. *Journal of Experimental Marine Biology and Ecology* **439**, 143-151 (2013).
- 16 MacDonald, J. A., Glover, T. & Weis, J. S. The impact of mangrove prop-root epibionts on juvenile reef fishes: a field experiment using artificial roots and epifauna. *Estuaries and Coasts* **31**, 981-993 (2008).
- 17 Verweij, M. *et al.* Structure, food and shade attract juvenile coral reef fish to mangrove and seagrass habitats: a field experiment. *Marine Ecology Progress Series* **306**, 257-268 (2006).
- 18 Enochs, I. C., Toth, L. T., Brandtneris, V. W., Afflerbach, J. C. & Manzello, D. P. Environmental determinants of motile cryptofauna on an eastern Pacific coral reef. *Marine Ecology Progress Series* **438**, 105-118 (2011).
- 19 Whitener, Z. T. & Nemeth, R. S. Effects of Site and Sampling Time on Motile Cryptic Invertebrate Communities on Fringing Reefs. (2013).
- 20 Wernberg, T., Thomsen, M. S., Staehr, P. A. & Pedersen, M. F. Epibiota communities of the introduced and indigenous macroalgal relatives *Sargassum muticum* and *Halidrys siliquosa* in Limfjorden (Denmark). *Helgoland marine research* **58**, 154-161 (2004).
- 21 Fabricius, K., De'ath, G., Noonan, S. & Uthicke, S. Ecological effects of ocean acidification and habitat complexity on reef-associated macroinvertebrate communities. *Proceedings of the Royal Society B: Biological Sciences* **281**, 20132479 (2014).
- 22 Enochs, I. & Manzello, D. Species richness of motile cryptofauna across a gradient of reef framework erosion. *Coral Reefs* **31**, 653-661 (2012).

- 23 Head, C. E. *et al.* Exceptional biodiversity of the cryptofaunal decapods in the Chagos Archipelago, central Indian Ocean. *Marine pollution bulletin* **135**, 636-647 (2018).
- 24 Allen, R., Foggo, A., Fabricius, K., Balistreri, A. & Hall-Spencer, J. M. Tropical CO₂ seeps reveal the impact of ocean acidification on coral reef invertebrate recruitment. *Marine pollution bulletin* **124**, 607-613 (2017).
- 25 Galimberti, A. *et al.* Integrated operational taxonomic units (IOTUs) in echolocating bats: a bridge between molecular and traditional taxonomy. *PLoS one* **7**, e40122 (2012).
- 26 Blaxter, M. L. The promise of a DNA taxonomy. *Philosophical Transactions of the Royal Society of London. Series B: Biological Sciences* **359**, 669-679 (2004).
- 27 Clare, E. L., Chain, F. J., Littlefair, J. E. & Cristescu, M. E. The effects of parameter choice on defining molecular operational taxonomic units and resulting ecological analyses of metabarcoding data. *Genome* **59**, 981-990 (2016).
- 28 Richardson, J. T. Eta squared and partial eta squared as measures of effect size in educational research. *Educational Research Review* **6**, 135-147 (2011).
- 29 Angelini, C. & Briggs, K. Spillover of secondary foundation species transforms community structure and accelerates decomposition in oak savannas. *Ecosystems*, 1-12 (2015).
- 30 Angelini, C. *et al.* Foundation species' overlap enhances biodiversity and multifunctionality from the patch to landscape scale in southeastern US salt marshes. *Proceedings of the Royal Society B: Biological Sciences* **282** (2015).
- 31 Angelini, C. & Silliman, B. R. Secondary foundation species as drivers of trophic and functional diversity: evidence from a tree-epiphyte system. *Ecology* **95**, 185-196 (2014).
- 32 Bishop, M. *et al.* Facilitation of molluscan assemblages in mangroves by the fucalean alga *Hormosira banksii*. *Marine Ecology Progress Series* **392**, 111-122 (2009).
- 33 Bishop, M. J., Byers, J. E., Marcek, B. J. & Gribben, P. E. Density-dependent facilitation cascades determine epifaunal community structure in temperate Australian mangroves. *Ecology* **93**, 1388-1401 (2012).
- 34 Bishop, M. J., Fraser, J. & Gribben, P. E. Morphological traits and density of foundation species modulate a facilitation cascade in Australian mangroves. *Ecology* **94**, 1927-1936 (2013).
- 35 Macreadie, P. I., Kimbro, D. L., Fourgerit, V., Leto, J. & Hughes, A. R. Effects of *Pinna* clams on benthic macrofauna and the possible implications of their removal from seagrass ecosystems. *Journal of Molluscan Studies* **80**, 102-106 (2014).
- 36 Siciliano, A., Schiel, D. R. & Thomsen, M. S. Effects of local anthropogenic stressors on a habitat cascade in an estuarine seagrass system. *Marine and Freshwater Research* **70**, 1129-1142 (2019).
- 37 Thomsen, M. S. *et al.* Modified kelp seasonality and invertebrate diversity where an invasive kelp co-occurs with native mussels. *Marine Biology* **165**, 173 (2018).
- 38 Thomsen, M. S. *et al.* Earthquake-driven destruction of an intertidal habitat cascade. *Aquatic Botany* **164**, 103217 (2020).
- 39 Thomsen, M. S., Metcalfe, I., South, P. & Schiel, D. R. A host-specific habitat former controls biodiversity across ecological transitions in a rocky intertidal facilitation cascade. *Marine and Freshwater Research* **67**, 144-152 (2016).
- 40 Thomsen, M. S. *et al.* Habitat cascades: the conceptual context and global relevance of facilitation cascades via habitat formation and modification. *Integrative and Comparative Biology* **50**, 158-175 (2010).
- 41 Thomsen, M. S. *et al.* Secondary foundation species enhance biodiversity. *Nature Ecology & Evolution* **2**, 634-639 (2018).
- 42 Borer, E. *et al.* What determines the strength of a trophic cascade? *Ecology* **86**, 528-537 (2005).
- 43 Gustafsson, C. & Salo, T. The effect of patch isolation on epifaunal colonization in two different seagrass ecosystems. *Marine biology* **159**, 1497-1507 (2012).
- 44 Bell, J. D., Steffe, A. S. & Westoby, M. Artificial seagrass: how useful is it for field experiments on fish and macroinvertebrates? *Journal of Experimental Marine Biology and Ecology* **90**, 171-177 (1985).
- 45 Johnson, M. W. & Heck Jr, K. L. Colonization and Predation in Isolated Seagrass Beds: An Experimental Assessment From the Northern Gulf of Mexico. *Gulf of Mexico Science* **26**, 1 (2008).

- 46 Yamada, K. & Kumagai, N. H. Importance of seagrass vegetation for habitat partitioning between
closely related species, mobile macrofauna *Neomysis* (Misidacea). *Hydrobiologia* **680**, 125-133
(2012).
- 47 Williams, B. S., Hughes, J. E. & Hunter-Thomson, K. Influence of epiphytic algal coverage on fish
predation rates in simulated eelgrass habitats. *The Biological Bulletin* **203**, 248-249 (2002).
- 48 Irving, A. D., Tanner, J. E. & McDonald, B. K. Priority effects on faunal assemblages within artificial
seagrass. *Journal of Experimental Marine Biology and Ecology* **340**, 40-49 (2007).
- 49 Hall, M. O. & Bell, S. S. Response of small motile epifauna to complexity of epiphytic algae on
seagrass blades. *Journal of Marine Research* **46**, 613-630 (1988).
- 50 Thomsen, M. S. *et al.* A meta-analysis of seaweed impacts on seagrasses: generalities and knowledge
gaps. *PloS one* **7**, e28595 (2012).
- 51 Höffle, H., Wernberg, T., Thomsen, M. S. & Holmer, M. Drift algae, an invasive snail and elevated
temperature reduce ecological performance of a warm-temperate seagrass, through additive
effects. *Marine Ecology Progress Series* **450**, 67-80 (2012).
- 52 Thomsen, M. S. *et al.* A sixth-level habitat cascade increases biodiversity in an intertidal estuary.
Ecology and evolution **6**, 8291-8303 (2016).
- 53 He, Q., Bertness, M. D. & Altieri, A. H. Global shifts towards positive species interactions with
increasing environmental stress. *Ecology letters* **16**, 695-706 (2013).
- 54 Gribben, P. E. *et al.* Facilitation cascades in marine ecosystems: a synthesis and future directions.
Oceanography and Marine Biology **57**, 127-168 (2019).
- 55 Lawton, J. H. Are there general laws in ecology? *Oikos*, 177-192 (1999).
- 56 Ravaglioli, C. *et al.* Positive cascading effects of epiphytes enhance the persistence of a habitat-
forming macroalga and the biodiversity of the associated invertebrate community under increasing
stress. *Journal of Ecology* **109**, 1078-1093 (2021).
- 57 Altieri, A. H. & Witman, J. D. Modular mobile foundation species as reservoirs of biodiversity.
Ecosphere **5**, 1-11 (2014).
- 58 Thomsen, M. S. *et al.* Impacts of marine invaders on biodiversity depend on trophic position and
functional similarity. *Marine Ecology Progress Series* **495**, 39-47 (2014).
- 59 Epstein, G. & Smale, D. A. *Undaria pinnatifida*: a case study to highlight challenges in marine invasion
ecology and management. *Ecology and evolution* **7**, 8624-8642 (2017).
- 60 South, P. M., Floerl, O., Forrest, B. M. & Thomsen, M. S. A review of three decades of research on the
invasive kelp *Undaria pinnatifida* in Australasia: An assessment of its success, impacts and status as
one of the world's worst invaders. *Marine Environmental Research* **131**, 243-257 (2017).
- 61 Thomsen, M. S., Wernberg, T., South, P. M. & Schiel, D. R. Non-native seaweeds drive changes in
marine coastal communities around the world, in 'Seaweed phylogeography' Springer. 147-185
(2016).
- 62 Robles, H. & Ciudad, C. Influence of habitat quality, population size, patch size, and connectivity on
patch-occupancy dynamics of the middle spotted woodpecker. *Conservation biology* **26**, 284-293
(2012).
- 63 Conner, R. N., Hooper, R. G., Crawford, H. S. & Mosby, H. S. Woodpecker nesting habitat in cut and
uncut woodlands in Virginia. *The Journal of Wildlife Management*, 144-150 (1975).
- 64 Cross, T. B., Latif, Q. S., Dudley, J. G. & Saab, V. A. Lewis's woodpecker nesting habitat suitability:
Predictive models for application within burned forests. *Biological Conservation* **253**, 108811 (2021).
- 65 Virkkala, R., Alanko, T., Laine, T. & Tiainen, J. Population contraction of the white-backed
woodpecker *Dendrocopos leucotos* in Finland as a consequence of habitat alteration. *Biological
conservation* **66**, 47-53 (1993).
- 66 Latif, Q. S., Saab, V. A., Dudley, J. G., Markus, A. & Mellen-McLean, K. Development and evaluation of
habitat suitability models for nesting white-headed woodpecker (*Dryobates albolarvatus*) in burned
forest. *PloS one* **15**, e0233043 (2020).
- 67 Barea, L. P. Nest-site selection by the Painted Honeyeater (*Grantiella picta*), a mistletoe specialist.
Emu-Austral Ornithology **108**, 213-220 (2008).
- 68 Watson, D. M. & Rawsthorne, J. Mistletoe specialist frugivores: latterday 'Johnny Appleseeds' or self-
serving market gardeners? *Oecologia* **172**, 925-932 (2013).

Reviewers' Comments:

Reviewer #1:

Remarks to the Author:

Comments to the reply of authors:

Amount as a measure of heterogeneity:

Reply: Here the authors responded to my critique on the use of amount as surrogate for heterogeneity by several arguments, e.g. definitions from Wikipedia and similar. Here I suggest staying in the field of ecological science and terminology. They also argue that in their meta-analyses Stein et al used several habitat-amount variables. Here I definitely disagree. Stein et al used only variables describing variation in environmental conditions, e.g. plant density as a surrogate for vegetation dimensions. This is true also for MacArthur's Foliage height diversity. Here, the focus is on an increase not of leaves per se but a more heterogeneous distribution of leaves in 3D space. Therefore, I highly recommend the authors to clarify in their manuscript if a variable describes amount or environmental heterogeneity.

Authors comment: We disagree that adaptations to heterogeneity only happen at the species level. Instead, we argue that adaptations can operate across all taxonomic resolutions. For example, virtually all small marine mobile organisms benefit from small interstitial spacing:

Reply: Here I disagree with the authors. Of course, specific environmental conditions filter for species with specific adaptations. Nevertheless, a species will evolve for a specific habitat because of better adaptations than relatives. This may happen independently in many lineages ending up with not related species of similar size in a specific environment.

Reply: I am happy to see additional analyses to account for different taxonomic resolutions.

Authors comment: This is of course correct, and we thank the reviewer for pointing it out – MacArthur did not measure biomass but used simpler non-destructive proxies.

Reply: Again, I disagree because complex forests often have not the highest biomass, even if higher growth rates. Therefore, the complexity measured by MacArthur is not a proxy of biomass.

Comments on the new manuscript:

Introduction:

I recommend starting a bit more general on the topic of environmental heterogeneity as determinant of biodiversity. General papers on the topic are still missing here, which would widen the scope of the manuscript (e.g. Allouche et al. 2012; Ben-Hur and Kadmon 2020).

Line 46/47: The first sentence can be removed. Go directly to the core theory.

Line 54/55 I highly suggest to talk about three environmental axis: habitat amount and two axis of heterogeneity.

Line 195-197: Here it would be much more interesting to see the contribution of amount and heterogeneity separately!

References

Allouche O, Kalyuzhny M, Moreno-Rueda G, Pizarro M, Kadmon R (2012) Area-heterogeneity tradeoff and the diversity of ecological communities. *Proc. Natl. Acad. Sci. U. S. A.* 109:17495-17500. doi: 10.1073/pnas.1208652109

Ben-Hur E, Kadmon R (2020) Heterogeneity-diversity relationships in sessile organisms: a unified framework. *Ecology Letters* 23:193-207. doi: 10.1111/ele.13418

Reviewer #2:

Remarks to the Author:

I commend the authors on the revision of the manuscript. I support the publication of the manuscript in its current form.

Reviewer #3:

Remarks to the Author:

I am satisfied that the authors have addressed all of my suggestions. Congratulations to the whole team on a fantastic paper.

Louise Firth

(Happy to reveal my identity)

Reply to reviewers comments, 26 Sep 2021

Reviewer 1: Amount as a measure of heterogeneity:

Here the authors responded to my critique on the use of amount as surrogate for heterogeneity by several arguments, e.g. definitions from Wikipedia and similar. Here I suggest staying in the field of ecological science and terminology.

- ***We appreciate the reviewer’s approach, but believe the arguments for our approach are consistent, have broad scientific support although perhaps more so among aquatic community ecologists, and is straightforward to interpret and understand. Consequently, we have not changed our approach. Instead, we have added text to the introduction, to highlight how and why we use the terminology the way we do. Inspired by Reviewer 1’s comments, we are now considering writing a small, more detailed, opinion piece about habitat-environment-heterogeneity-complexity terminology that could be constructive for similar future discussions. Below we repeat our arguments for why different amounts of habitat is an example of habitat-heterogeneity.***
- ***First, we note that there are more than 100 scientific usages of terminology related to environmental and habitat heterogeneity and complexity (Stein et al. 2014), highlighting the vast variation, and likely disagreement, among scientists about how to define and use these terms. It is virtually impossible to follow a specific advice about ‘staying in the field of ecological science and terminology’ because there is no consensus nor single scientific definition or use of ‘heterogeneity’.***
- ***For example, Stein et al (2004) - a paper that focused on environmental, not habitat, heterogeneity - does not define the terminology but simply states that “Here, we use environmental heterogeneity as an umbrella term for all terms relating to spatial complexity, diversity, heterogeneity, or structure in the environment.” Under this umbrella approach, the amount of habitat (i.e., the biomass of foundation species in a specific area that is inhabited by small invertebrates) is a straightforward example of ‘all terms related to the structure in the environment’.***
- ***Importantly, Tokeshi & Arakaki's (2012) insightful review paper used heterogeneity and complexity as synonyms and described in detail the importance of both sizes and density of structural elements to understand relationships between biodiversity and habitat heterogeneity. In other words, Tokeshi's & Araki's sizes and abundances of structural elements is equivalent to our biomass of foundation species.***
- ***Similarly, Tews et al. (2004), in their classic heterogeneity paper, also argue that more and taller biogenic structures ('keystones', i.e., amounts of a specific biogenic habitat), like large trees on savannas, increase an areas habitat heterogeneity. Specifically, Table 2 in Tews et al. list vegetation height, vegetation cover and***

structural extent as traditional heterogeneity measures. Their broad usage of habitat heterogeneity also aligns closely with our more specific analysis of animal diversity associated with secondary foundation species - like epiphytes attached to seaweed, seagrass or trees, bivalves embedded within mussels, or oysters attached to mangroves. Thus, Tews et al.'s. height, cover and extent of keystone structures is directly analogous to how we quantify and analyse biomass of secondary foundation species.

- *Furthermore, when a terminology is scientifically vague, ambiguous, has different meanings for different scientist and lacks consensus, we advocate that scientists instead revisit first principles, such as definitions from dictionaries like Wikipedia, Oxford, online dictionary, and the online biological dictionary (the latter being relatively 'scientific'). These dictionaries all define heterogeneity in a context of non-uniformity in traits, characteristics, and objects (i.e., differences), with a broad approach. Therefore; (a) 'different amount' is a form of heterogeneity and (b) 'amount or biomass of foundation species' is a type of habitat that is inhabited by animals.*
- *Finally, we highlight – again – that our 'amount' trait is measured within a specific similar sized area for each of our 22 individual experiments. Our data collection is therefore NOT equivalent to accumulated species-area or species-sample data – that sometimes are distinguished from habitat heterogeneity (Nally and Watson 1997, Silva et al. 2018).*

Reviewer 1. They also argue that in their meta-analyses Stein et al used several habitat-amount variables. Here I definitely disagree. Stein et al used only variables describing variation in environmental conditions, e.g. plant density as a surrogate for vegetation dimensions.

- *We don't understand; Although Stein used CV, SD and other variability/diversity metrics for some variables, they also include percent cover of rocks, grass and forests, mean patch size and plant density – variables that all correlate with amounts of habitats (for those species with high affinities for rocks, grass, forests, patches and plants, respectively) – see table 1 below. In other words, Stein et al included habitat-amount variables like we did (and like Tews et al did in Table 2).*

Table 1 Subject areas of environmental heterogeneity (EH) categorised into EH measure categories relating to the same concepts with example measures used for quantification of EH

EH subject area	EH measure category	Example measures
Land cover	Land cover proportion	% Cover of forest; % cover of grassland
	Land cover diversity	# Land cover types; Shannon index of land cover types
Vegetation	Patchiness	Edge density; mean patch size
	Plant diversity	# Plant species; Shannon index of tree species
	Vegetation complexity	Foliage height diversity; PCA of vegetation variables
Climate	Vegetation dimension	CV of trunk perimeter; density of plants
	Climate	CV of precipitation; temperature range
Soil	Soil diversity	# Soil types; Shannon index of soil types
	Soil variables	CV of soil moisture; SD of soil pH
Topography	Elevation diversity	Elevation range; SD of elevation
	Microtopography	# Microtopographic elements; % cover of rocks
	Profile	SD of profile curvature; slope
Mixed	Mixed	Composite heterogeneity index; # ecological variables present

CV, coefficient of variation; #, number of; %, percentage of; PCA, principal component analysis; SD, standard deviation.

Table 1 in Stein et al. Measures that correlate with amounts of habitat are highlighted in yellow.

Table 2 Measurement of habitat heterogeneity.

Study sites	Discrete variables (structural elements)		Continuous variables (structural qualities)	
	Single	Multiple	Single	Multiple
Definition	Number of structural elements	Number and evenness of structural elements	Extent of structural qualities	Structural difference between various sites
Name	Structural richness	Structural diversity	Structural extent	Structural gradient
Measurement	Count of the elements	Shannon's index of diversity	Measured structural quality	Gradient length, Euclidian distances
Example	Number of habitat types in a landscape	Diversity of habitat types in a landscape	Vegetation height or coverage	Difference in vegetation structure between sites

Table 2 in Tews et al. Measures that correlate with amounts of biogenic habitat are highlighted in yellow.

Reviewer 1. This is true also for MacArthur's Foliage height diversity. Here, the focus is on an increase not of leaves per se but a more heterogeneous distribution of leaves in 3D space.

- ***Again, we don't understand; MacArthur did not measure 'heterogeneous distribution of leaves in 3D space'. MacArthur measured the height and width of trees (see Fig. 1 in MacArthur inserted below) – two traditional types of amounts - to calculate a derived heterogeneity/diversity metric. For example, graphical analyses of Figure 1 in MacArthur show higher bird diversity associated with high amounts of tree-habitat in Maryland and Florida compared to low amounts of tree-habitat in Vermont and Penn (J,I are tall and wide compared to short and narrow A,B). Again, like MacArthur, we included a variable (the biomass of a structural element) that is fundamentally/allometrically correlated with an element's width and height.***

FIG. 1. The densities of foliage (measured in square feet of leaf silhouette per cubic foot of space) are plotted along the abscissae. The height in feet above the ground is the ordinate. F.H.D. is foliage height diversity, B.S.D. is bird species diversity, and P.S.D. is plant species diversity.

Figure 1 in MacArthur showing heights and width of tree strata (which correlate with bird diversity = BSD).

Reviewer 1. Therefore, I highly recommend the authors to clarify in their manuscript if a variable describes amount or environmental heterogeneity.

1. **Based on scientific rationale and key papers discussed above (Tews et al. 2004, Tokeshi and Arakaki 2012) - and references therein - that support our definition of habitat heterogeneity, we decided to keep different 'amounts of habitat' within a specific area, as an example of a habitat-heterogeneity axis. In particular, we like that these approaches align with first-principal dictionary definitions (which we argue are preferable to ambiguous and conflicting scientific usages). However, to avoid misunderstanding with researchers that may disagree with Tokeshi and Arakaki and Tews et al., we added a few lines to the manuscript that state our approach explicitly.**

Review 1. Comments on the new manuscript:

Introduction: I recommend starting a bit more general on the topic of environmental heterogeneity as determinant of biodiversity. General papers on the topic are still missing here, which would widen the scope of the manuscript (e.g. Allouche et al. 2012; Ben-Hur and Kadmon 2020).

Done. This is a good suggestion. We have added text about heterogeneity to capture the essence of these two key references and increase the generality of our study before we introduce our 3 specific test-factors in more detail. The added text reads... “Although most studies have shown positive relationships between heterogeneity and diversity, certain processes, like effective area per species, can result in negative relationships between heterogeneity and biodiversity (Allouche et al. 2012, Ben-Hur and Kadmon 2020)”. **Note that our paper is mainly about habitat-heterogeneity - a more specific topic than environmental heterogeneity and that we had already cited Ben-Hur & Kadmon 2020 (our previous reference 20).**

Line 46/47: The first sentence can be removed. Go directly to the core theory.

Done.

Line 54/55 I highly suggest to talk about three environmental axis: habitat amount and two axis of heterogeneity.

We strongly disagree. It is unnecessarily complicated and cumbersome to explain if and how habitat relates to the environment and what heterogeneity is in a short abstract. Instead, we are certain that readers of Nature Communication know that habitat is part the environment, habitat is the place where organisms live, and heterogeneity reflects non-uniformity. The sentence in question simply says “three axes of habitat-heterogeneity” concisely reflecting what we did (i.e., we tested for impacts of (a) amount of habitat, (b) complexity of habitat and (c) function of habitat on biodiversity).

Line 195-197: Here it would be much more interesting to see the contribution of amount and heterogeneity separately!

We strongly disagree. There is no reason to statistically evaluate or prioritize ‘amount’ over ‘function’ or ‘complexity’ in line 195-197. Most importantly, in line 159-193 we already evaluated the contribution of amount vs. other types of heterogeneity (and their interactions) in great detail – describing, comparing, and contrasting >30 Sum of Squares values. Line 195-197 simply summarized all of these SS-results to highlight that our controlled test-factors (axes of heterogeneity) explained much more data variability than the unexplained spatiotemporal covariates (which were difficult to control due to massive variation associated with doing global experiments in vastly different ecosystems). In other words, the specific sentence nails down that the universal habitat attributes were much more important than spatiotemporal background conditions.

References

Allouche O, Kalyuzhny M, Moreno-Rueda G, Pizarro M, Kadmon R (2012) (Allouche et al. 2012). Proc. Natl. Acad. Sci. U. S. A. 109:17495-17500. doi: 10.1073/pnas.1208652109

Ben-Hur E, Kadmon R (2020) Heterogeneity-diversity relationships in sessile organisms: a unified framework. Ecology Letters 23:193-207. doi: 10.1111/ele.13418

Nally, R. M., and D. M. Watson. 1997. Distinguishing area and habitat heterogeneity effects on species richness: birds in Victorian buloke remnants. Australian Journal of Ecology 22:227-232.

- Silva, J. B., J. C. Sfair, N. D. d. Santos, and K. C. Pôrto. 2018. Bryophyte richness of soil islands on rocky outcrops is not driven by island size or habitat heterogeneity. *Acta Botanica Brasílica* **32**:161-168.
- Stein, A., K. Gerstner, and H. Kreft. 2014. Environmental heterogeneity as a universal driver of species richness across taxa, biomes and spatial scales. *Ecology letters* **17**:866-880.
- Tews, J., U. Brose, V. Grimm, K. Tielbörger, M. C. Wichmann, M. Schwager, and F. Jeltsch. 2004. Animal species diversity driven by habitat heterogeneity/diversity: the importance of keystone structures. *Journal of Biogeography* **31**:79-92.
- Tokeshi, M., and S. Arakaki. 2012. Habitat complexity in aquatic systems: fractals and beyond. *Hydrobiologia* **685**:27-47.

Reviewer #2 (Remarks to the Author):

I commend the authors on the revision of the manuscript. I support the publication of the manuscript in its current form.

Thank you.

Reviewer #3 (Remarks to the Author):

I am satisfied that the authors have addressed all of my suggestions. Congratulations to the whole team on a fantastic paper. Louise Firth. (Happy to reveal my identity)

Thank you.

Reviewers' Comments:

Reviewer #1:

Remarks to the Author:

First I apologize for delay. I would like to thank the authors for the intensive discussion around my points of criticism. I can accept the point of view of different amounts as a form of heterogeneity as it is presented now. Only with regard to MacArthur, I would like to take the liberty of noting that not simply the height and width of trees but the distribution of vegetation in space was calculated as foliage height diversity and correlated with bird diversity.

Point-by-point reply to reviewers comment

Reviewer 2 and 3: No comments – manuscript accepted.

Reviewer 1 comment: I can accept the point of view of different amounts as a form of heterogeneity as it is presented now. Only with regard to MacArthur, I would like to take the liberty of noting that not simply the height and width of trees but the distribution of vegetation in space was calculated as foliage height diversity and correlated with bird diversity.

Reply. Done. We have changed the sentence to reflect the reviewer's comment (see red addition). The sentence now say: "For example, seminal work by MacArthur and MacArthur demonstrated how bird diversity increases with 'foliage height diversity' (diversity of elements), **an index that varies with the distribution of vegetation in space**, including the width and height of vegetation strata (sizes of elements)."